# Structure of Vps4 with circular peptides and implications for translocation of two polypeptide chains by AAA+ ATPases

Han Han[1], James M Fulcher[1], Venkata P Dandey[2], Janet H Iwasa[1], Wesley I Sundquist[1], Michael S Kay[1], Peter S Shen[1]*, Christopher P Hill[1]*

[1]Department of Biochemistry, University of Utah, Salt Lake City, United States; [2]Simons Electron Microscopy Center, New York Structural Biology Center, New York, United States

**Abstract** Many AAA+ ATPases form hexamers that unfold protein substrates by translocating them through their central pore. Multiple structures have shown how a helical assembly of subunits binds a single strand of substrate, and indicate that translocation results from the ATP-driven movement of subunits from one end of the helical assembly to the other end. To understand how more complex substrates are bound and translocated, we demonstrated that linear and cyclic versions of peptides bind to the *S. cerevisiae* AAA+ ATPase Vps4 with similar affinities, and determined cryo-EM structures of cyclic peptide complexes. The peptides bind in a hairpin conformation, with one primary strand equivalent to the single chain peptide ligands, while the second strand returns through the translocation pore without making intimate contacts with Vps4. These observations indicate a general mechanism by which AAA+ ATPases may translocate a variety of substrates that include extended chains, hairpins, and crosslinked polypeptide chains.
DOI: https://doi.org/10.7554/eLife.44071.001

**\*For correspondence:**
peter.shen@biochem.utah.edu
(PSS);
chris@biochem.utah.edu (CPH)

## Introduction

The large and diverse family of AAA+ ATPases (ATPases Associated with various Activities) (*Erzberger and Berger, 2006*) includes multiple members that form hexamers and are believed to unfold protein substrates by translocating them through their central pore (*Nyquist and Martin, 2014*). The structures of several AAA+ ATPases have been determined by electron cryo-microscopy (cryo-EM) in the presence of engaged substrate, including the mitochondrial inner membrane protease YME1 (*Puchades et al., 2017*), the disaggregase Hsp104 (*Gates et al., 2017*), the chaperone ClpB (*Deville et al., 2017*; *Yu et al., 2018*), the TRIP13 mitotic checkpoint regulator (*Alfieri et al., 2018*), the SNARE disassembly machine NSF (*White et al., 2018*), the VAT archaeal homolog of Cdc48/p97 (*Ripstein et al., 2017*), the proteasome (*de la Peña et al., 2018*; *Dong et al., 2019*), and Vps4 (*Han et al., 2017*; *Monroe et al., 2017*), which associates with the positive regulator Vta1 (*Azmi et al., 2006*; *Lottridge et al., 2006*; *Scott et al., 2005*) to drive the ESCRT pathways that mediate multiple membrane fission events in eukaryotic cells by disassembling filaments comprised of ESCRT-III subunits (*McCullough et al., 2018*). Structures of all of these complexes have been determined in an asymmetric, lock-washer conformation in which four or five of the six subunits in the hexamer form a helix that adopts a right-handed helical configuration, while the other one or two subunits are displaced from the helical axis, as if transitioning between ends of the helix.

These AAA+ ATPase complexes bind the substrate polypeptide in the central pore in an extended conformation, which in the case of Vps4 has been modeled as a β-strand-like conformation whose right-handed helical symmetry (60° rotation and ~6.5 Å displacement every two amino acid residues) matches the symmetry of the helical AAA+ ATPase subunits (*Han et al., 2017*;

*Monroe et al., 2017*). Although the resolution of currently available AAA+ ATPase substrate complexes makes it challenging to model precise details of the substrate structure, this conformation is appealing because it allows the substrate to bind the helical AAA+ ATPase subunits with successive dipeptides of the substrate making equivalent interactions with the enzyme and because it is accessible for almost all amino acid residues. Some variations from the canonical conformation are likely to occur, especially for sequences that contain proline, which has a fixed −60° phi angle, and glycine, which is flexible and lacks a side chain, which seems to be important for binding. Interfaces between the helical AAA+ ATPase subunits are stabilized by binding of ATP at the active site of the first subunit and contacts with the 'finger' arginine residues of the following subunit. These observations suggest a model in which ATP hydrolysis at the last interface in the helix promotes disengagement to an open, transitioning conformation that allows nucleotide exchange, with ATP binding to the transitioning subunit allowing it to rejoin the growing end of the helical assembly and bind the next dipeptide of the extended substrate polypeptide. Similar structures for multiple AAA+ ATPase peptide complexes (*Alfieri et al., 2018*; *de la Peña et al., 2018*; *Deville et al., 2017*; *Dong et al., 2019*; *Gates et al., 2017*; *Han et al., 2017*; *Monroe et al., 2017*; *Puchades et al., 2017*; *Ripstein et al., 2017*; *White et al., 2018*) support a general model in which cycles of this process cause the AAA+ ATPase to 'walk' along its substrate and thereby translocate the substrate through its central hexameric pore (*Han and Hill, 2019*).

This model explains how an extended polypeptide substrate might be translocated, but it does not explain the translocation of more complex substrates. For example, the proteasome can process substrates starting from internal loops (*Kraut and Matouschek, 2011*), substrates that are cross-linked (*Lee et al., 2002*), and substrates that are conjugated to ubiquitin (*Shabek and Ciechanover, 2010*). Similarly, Cdc48 can process substrates that are covalently ligated to ubiquitin chains (*Bodnar and Rapoport, 2017*), and ClpXP can process disulfide-cross-linked dimers (*Burton et al., 2001*).

Here, we show that the mechanism proposed for linear, extended polypeptides is also compatible with translocation of more complex substrates. Peptides that include a known Vps4-binding sequence were synthesized in linear and circular configurations and shown to bind Vps4 with similar affinities. Structure determination showed that a primary segment of the circular peptide binds indistinguishably from the isolated linear peptide, while a secondary segment packs against it in a β-ladder hairpin configuration that passes through the hexamer pore without distorting the Vps4 structure or making intimate contacts with Vps4. These observations indicate that AAA+ ATPases can translocate two chains of a substrate, such as would occur in crosslinked chains or in ubiquitin conjugates, using the same mechanism as for an extended polypeptide.

## Results and discussion

### Linear and circular peptides bind Vps4 with similar affinity

The experimental design was guided by our previously reported biochemical studies (*Han et al., 2015*; *Monroe et al., 2014*) and cryo-EM structure (*Han et al., 2017*; *Monroe et al., 2017*) of Vps4 in complex with an 8-residue peptide (DEIVNKVL; peptide F) that was derived from the yeast ESCRT-III subunit, Vps2 (*Han et al., 2015*). Although this peptide was originally discovered as a relatively tight-binding sequence, we subsequently found that its binding affinity is comparable to a diverse range of peptide sequences (data not shown). This indicates that its complex with Vps4 reflects a canonical translocating state, as does its structural similarity with multiple other AAA+ ATPase complexes (*Han and Hill, 2019*). ADP·BeF$_x$ was used as the non-hydrolysable ATP analog because our earlier studies indicated that it stabilizes the Vps4 hexamer and supports peptide binding to a greater extent than AMPPNP or ATPγS, presumably because it is a better mimic of ATP at the Vps4 active site (*Han et al., 2015*).

The following peptides were synthesized using Fmoc solid-phase peptide synthesis with acetylated N-termini and amidated C-termini: F12 (peptide F flanked on both ends by two glycine residues); F30 (peptide F extended by four residues at the N-terminus and 18 residues at the C-terminus; FF30 (F30 but including a second copy of peptide F) (*Figure 1*, *Figure 1—figure supplements 1–9*). Most of the additional residues in F30 and FF30 were glycine, alanine, or serine, which are not expected to bind strongly to Vps4 (*Han et al., 2017*). Lysine was included at position

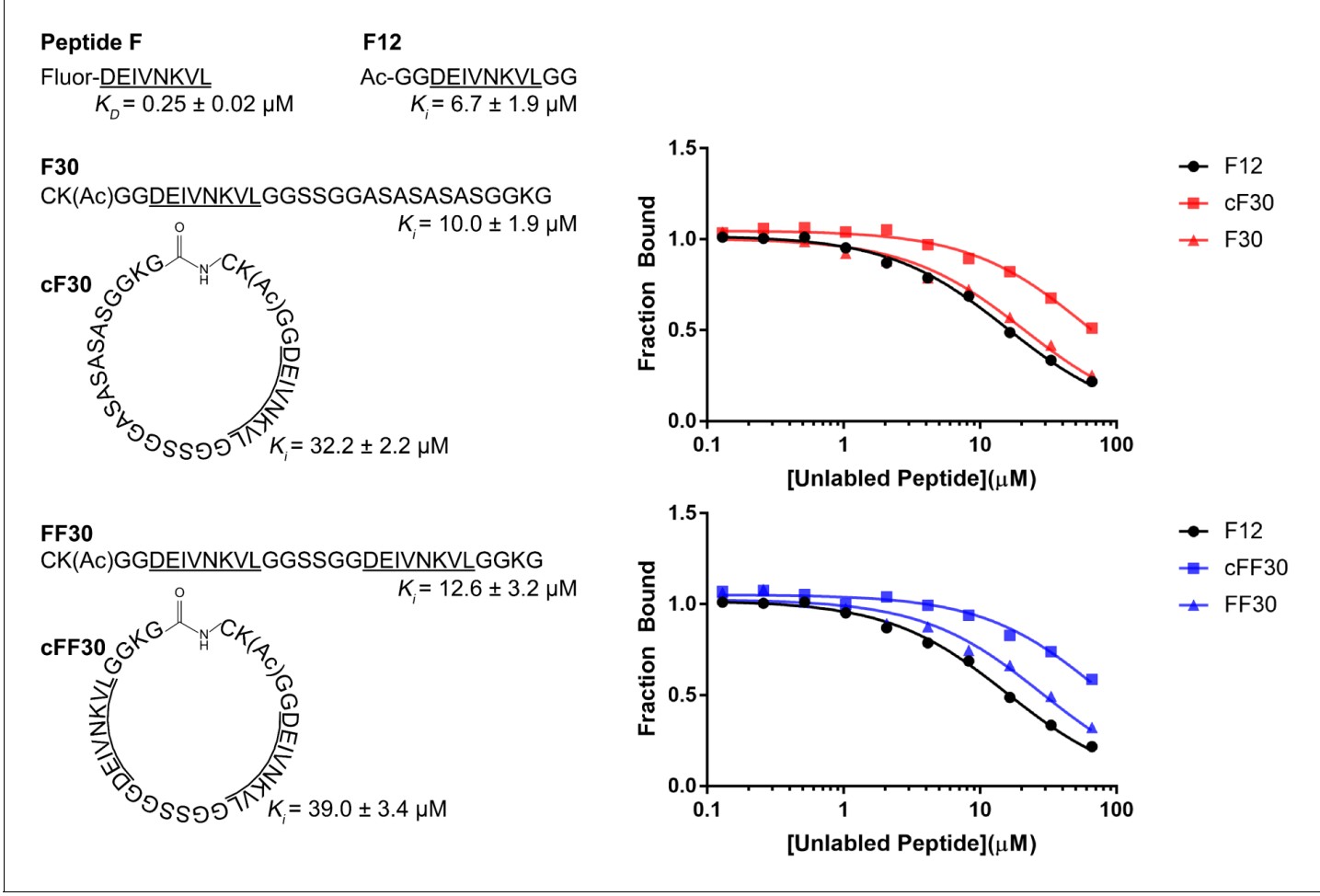

**Figure 1.** Linear and circular peptides bind Vps4 with similar affinities. Sequences of the linear and circular peptides used in this study are shown, together with competition fluorescence polarization binding isotherms and calculated $K_D$ and $K_i$ values.

DOI: https://doi.org/10.7554/eLife.44071.002

The following source data and figure supplements are available for figure 1:

**Source data 1.** Binding of peptides to Vps4 in a fluorescence polarization competition assay related to *Figure 1* and *Figure 1—figure supplement 1*.
DOI: https://doi.org/10.7554/eLife.44071.013
**Figure supplement 1.** Competitive binding assays.
DOI: https://doi.org/10.7554/eLife.44071.003
**Figure supplement 2.** Characterization of HPLC purified peptide F by LC-MS.
DOI: https://doi.org/10.7554/eLife.44071.005
**Figure supplement 3.** Characterization of HPLC purified F12 by LC-MS.
DOI: https://doi.org/10.7554/eLife.44071.006
**Figure supplement 4.** Characterization of HPLC purified F30 by LC-MS.
DOI: https://doi.org/10.7554/eLife.44071.007
**Figure supplement 5.** Characterization of HPLC purified cF30 by LC-MS.
DOI: https://doi.org/10.7554/eLife.44071.008
**Figure supplement 6.** Characterization of HPLC purified FF30 by LC-MS.
DOI: https://doi.org/10.7554/eLife.44071.009
**Figure supplement 7.** Characterization of HPLC purified cFF30 by LC-MS.
DOI: https://doi.org/10.7554/eLife.44071.010
**Figure supplement 8.** High-resolution LC-MS analysis of trypsin-digested peptides.
DOI: https://doi.org/10.7554/eLife.44071.011
**Figure supplement 9.** Comparison of MS2 spectra from trypsin-digested linear and cyclic peptides.
DOI: https://doi.org/10.7554/eLife.44071.012

2 to allow labeling (not used in this study) and at position 29 to promote solubility. An N-terminal cysteine was included in F30 and FF30, and versions of these peptides were also synthesized with a C-terminal hydrazide to facilitate synthesis of the circular cF30 and cFF30 peptides, which are identical to F30 and FF30 except for cyclization through a peptide bond between the N and C terminal residues (*Figure 1*).

Competitive fluorescence polarization showed that F12, F30, FF30, cF30, and cFF30 all bound Vps4 with similar affinities, with the cyclized peptide cF30 and cFF30 showing slightly weaker binding (~3 fold) (*Figure 1*). Essentially identical binding constants were determined for binding to Vps4 and to Vps4-Hcp1 (*Figure 1—figure supplement 1*), which is the stable hexamer construct used for structural studies. These data indicate that linear and circular versions of the same peptide bind Vps4 with similar affinities, and that structure determination with the Hcp1 fusion will provide a good representation of the association with the isolated Vps4 AAA+ ATPase cassette.

## Structure of Vps4-circular peptide complexes

We determined cryo-EM structures of cF30 and cFF30 complexes using the same approach as for the previously reported linear peptide complex (*Han et al., 2017*; *Monroe et al., 2017*). Constructs of Vps4-Hcp1 and the VSL domain of the activator protein Vta1 were the same as the earlier studies, as were the concentrations of Vps4-Hcp1, ADP·BeF$_x$ and peptide, and the glutaraldehyde crosslinking procedures. The only difference was a 10-fold higher concentration of Vta1$^{VSL}$, which was increased because the earlier cryo-EM reconstructions showed low Vta1$^{VSL}$ occupancy.

Density maps were reconstructed for the cF30 and cFF30 complexes at 3.8 Å and 4.0 Å resolution, respectively (*Figure 2*, *Figure 2—figure supplements 1–8*, *Figure 2—videos 1–3*, *Table 1*). No differences are apparent in the refined models, except that the Vta1$^{VSL}$ cofactor protein is better defined in the cFF30 complex structure, probably because Spotiton (*Dandey et al., 2018*) was used to prepare the cFF30 grids (below). Because other regions of the cF30 and cFF30 reconstructions are essentially identical, we used particles from both datasets to reconstruct a combined map at 3.6 Å. The overall structure superimposes closely with the previously reported linear peptide complex (*Han et al., 2017*), including the same helical arrangement of five Vps4 subunits (subunits A-E). Although details of nucleotide configuration are not definitively resolved, consistent with the earlier structure, the subunit AB, BC, and CD interfaces appear to bind ADP·BeF$_x$ (ATP), while the DE interface density is ambiguous, but could be ADP or an ADP/ADP·BeF$_x$ mixture (*Figure 2C*, *Figure 2—video 3*). Density at the subunit E active site is consistent with binding to ADP, and the subunit F active site has such weak density that it does not indicate whether or not nucleotide is bound.

## Circular peptides bind the Vps4 pore in a hairpin conformation

The density and refined models for the cF30 and cFF30 peptides are essentially identical (*Figure 2D*), with two polypeptide strands passing through the Vps4 pore (*Figure 3A*, *Figure 3—video 1*). The pore loop positions and the substrate strand with the strongest density (primary strand) superimpose closely on the earlier peptide F complex structure, except that density for the peptide now extends both N-terminally and C-terminally for two additional residues (*Figure 3B*, *Figure 3—video 2*). The density is consistent with the peptide F sequence binding in the same register as the earlier peptide F complex, with odd-numbered residues binding to an array of class I pockets and even-numbered residues binding to an array of class II pockets (*Han et al., 2017*) (*Figure 3C*, *Figure 3—video 3*). We attempted to discern the orientation of this peptide strand by comparing the assigned orientation with a peptide model built and refined optimally in the reverse orientation (*Figure 2—figure supplement 4*). This analysis showed that the map-model correlation coefficients and the EMRinger scores (*Barad et al., 2015*) slightly favor the assigned orientation, but are not definitive. This ambiguity is expected for the current 3.6 Å resolution, and the peptide orientation remains an important question for future studies.

The returning strand of the circular peptides has weaker density, indicating that it is more mobile (*Figure 2D*). Density for these side chains is not strongly defined, although the sequence of the cFF30 peptide and the presence of two residues on either side of the primary strand F-peptide motif means that at least 5 of the returning strand residues in the pore region must have relatively large side chains. Nevertheless, the density indicates that the returning strand adopts an extended conformation in which eight residues are reasonably modeled as forming a β-ladder interaction with the

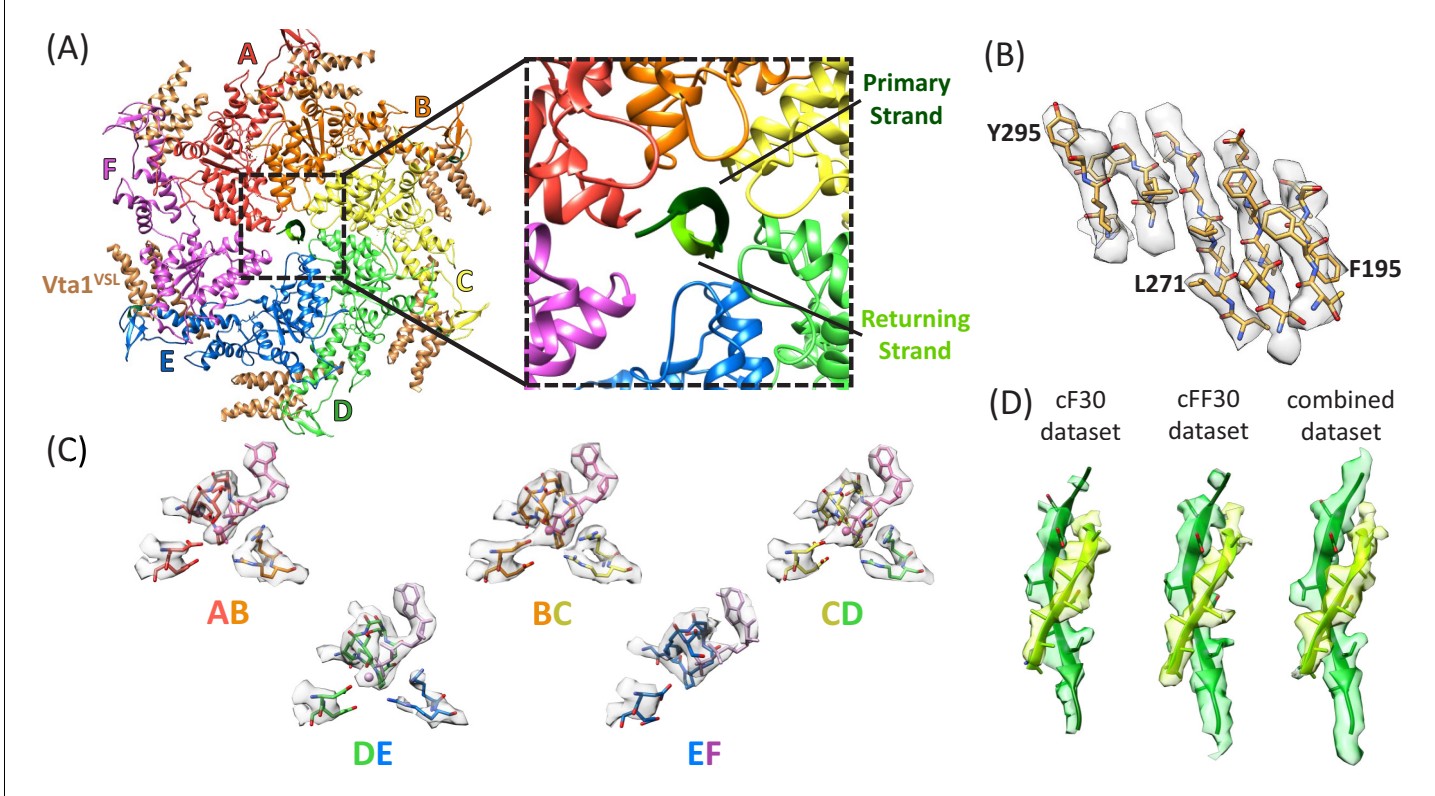

**Figure 2.** Structure determination. (A) Overall structure of the Vps4-cyclic peptide complex. The close up view of the pore region shows the primary strand (dark green) and returning strand (light green) of the cyclic peptide. (B) Representative section of density in the large domain of the Vps4 B subunit. (C) Density around the nucleotides and coordinating residues for the active sites of Vps4 subunits A-E. These binding sites occur at the interface with the following subunit. (D) Density around the circular peptides. Shown as side views separately for the cF30, cFF30, and combined maps.
DOI: https://doi.org/10.7554/eLife.44071.014

The following video and figure supplements are available for figure 2:

**Figure supplement 1.** Cryo-EM structure determination and validation of Vps4 bound to cyclic peptides.
DOI: https://doi.org/10.7554/eLife.44071.015

**Figure supplement 2.** Cryo-EM processing workflow of the Vps4-cF30 complex.
DOI: https://doi.org/10.7554/eLife.44071.016

**Figure supplement 3.** Cryo-EM processing workflow of Vps4-cFF30 complex.
DOI: https://doi.org/10.7554/eLife.44071.017

**Figure supplement 4.** Data processing workflow for combining the cyclic peptide datasets and subunit F classification.
DOI: https://doi.org/10.7554/eLife.44071.018

**Figure supplement 5.** Validation of subunit F classes.
DOI: https://doi.org/10.7554/eLife.44071.019

**Figure supplement 6.** The three subunit F conformations.
DOI: https://doi.org/10.7554/eLife.44071.020

**Figure supplement 7.** Fit of peptide to the cF30/cFF30 combined density when refined in the assigned and reversed orientations.
DOI: https://doi.org/10.7554/eLife.44071.021

**Figure supplement 8.** Focused 3D classification of Vta1$^{VSL}$.
DOI: https://doi.org/10.7554/eLife.44071.022

**Figure 2—video 1.** Representative density shown around the β-sheet of Vps4 subunit B for the combined cF30 and cFF30 dataset.
DOI: https://doi.org/10.7554/eLife.44071.023

**Figure 2—video 2.** Density around the cyclic peptide.
DOI: https://doi.org/10.7554/eLife.44071.024

**Figure 2—video 3.** Density around the nucleotides.
DOI: https://doi.org/10.7554/eLife.44071.025

**Table 1.** Reconstruction, refinement, and validation statistics.

| Reconstruction | |
| --- | --- |
| Number of particle images | 237,480 |
| Resolution (0.143 FSC) (Å) | 3.6 |
| Map sharpening B-factor (Å$^2$) | −157 |
| EMDB accession number | EMD-0443 |
| **Model refinement of Vps4 subunits A-E** | |
| PDB accession number | 6NDY |
| Resolution used for refinement (Å) | 3.6 |
| Number of atoms | 12531 |
| RMSD: Bond length (Å) | 0.003 |
| RMSD: Bond angles (°) | 0.739 |
| Ramachandran: Favored (%) | 94.1 |
| Ramachandran: Allowed (%) | 5.9 |
| Ramachandran: Outlier (%) | 0 |
| **Validation** | |
| Molprobity score/percentile (%) | 1.64 (100%) |
| Clashscore/percentile (%) | 4.67 (100%) |
| EMRinger score | 1.04 |

DOI: https://doi.org/10.7554/eLife.44071.026

primary strand. The 10 residues of the circular peptides that lack experimental density can be reasonably modeled in hairpin turn conformations (*Figure 3A*).

## Insights into substrate translocation

The second strand of the circular peptide is accommodated within the Vps4 pore by the displacement of subunit F from the substrate-binding groove and by the peptide adopting the same helical symmetry as Vps4 subunits A-E. The pseudo two-fold axis along the length of the circular peptide β-ladder aligns with the helical axis of Vps4 subunits A-E (*Figure 4*, *Figure 4—video 1*), thereby maximizing the distance of the second strand away from the helical Vps4 subunits (A-E) that bind the primary strand, and maximizing the space available for the second peptide strand. An open question is whether or not it is possible to accommodate a third strand without distorting the Vps4 structure.

The hinge angle between the large and small ATPase domains (*Gonciarz et al., 2008*) varies by just 2° (126–128°) for subunits A-E, but is more open and variable (126–141°) for the three subunit F models derived from focused classification of the combined cF30 and cFF30 particles (*Figure 5A*). The more variable and open subunit F structure, and the lack of close contacts between the large ATPase domain of subunit F and the large ATPase domains of its neighboring subunits A and E (*Figure 5B*), is consistent with nucleotide exchange occurring during transit from the subunit-E state to the subunit-A state.

The proposed ~30 Å transition of subunit F from the subunit-E end of the Vps4 helix to the subunit-A end of the helix is consistent with the results of focused classification of the various Vps4 datasets, including those of cF30, cFF30, combined cF30 + cFF30, and the two linear peptide structures (*Han et al., 2017*; *Monroe et al., 2017*). These classifications each provide two or three maps that show distinct positions for subunit F, all of which avoid contact with the returning strand of the cyclic peptides and together span the path traversed during cycle of the proposed translocation mechanism (*Figure 5C*, *Figure 5—video 1*).

The focused classification may indicate a mechanism to trigger ATP hydrolysis preferentially at the subunit D active site (DE interface), which is thought to give directionality the Vps4 translocation cycle (*Han et al., 2017*; *Monroe et al., 2017*). Specifically, the subunit F state closest to docking against subunit A (F$_3$), correlates with the subunit E small domain rotating 7° and shifting 3 Å relative to the primary position that is seen in all other reconstructions. Thus, the final stage of the subunit F

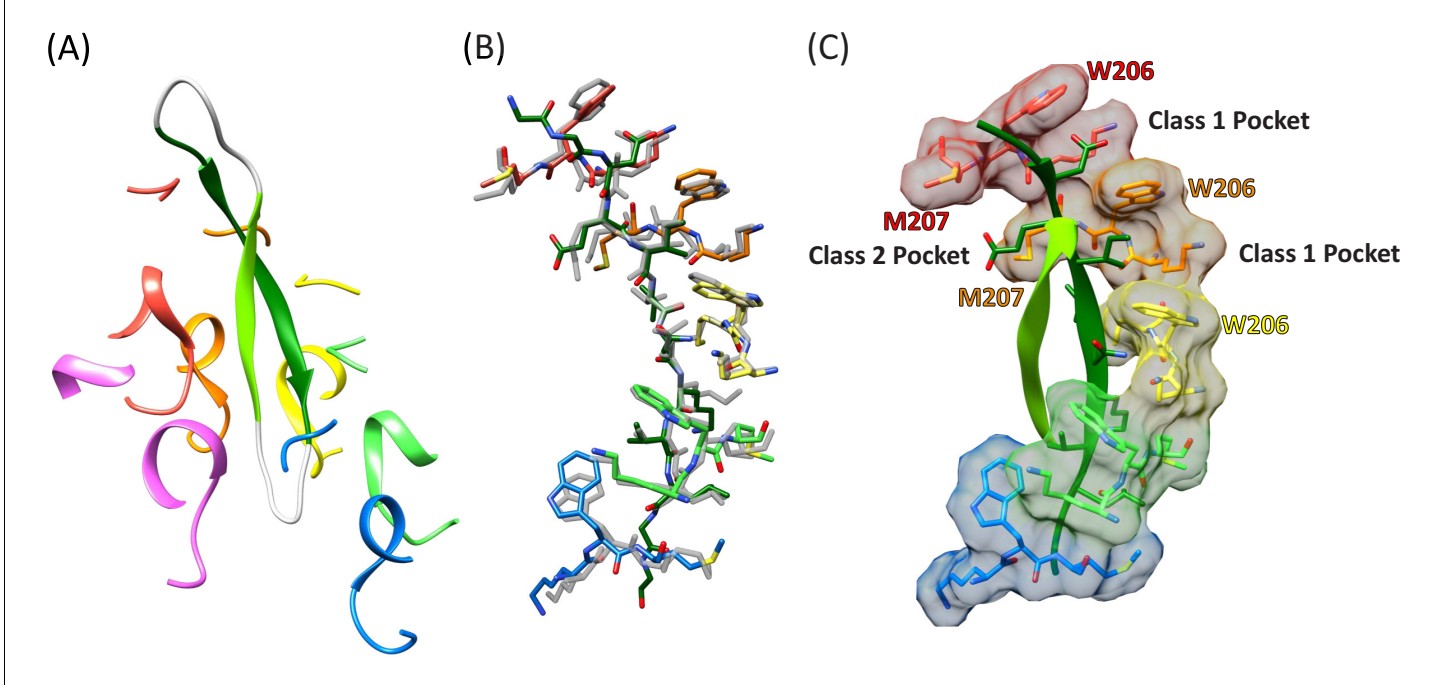

**Figure 3.** Cyclic peptide structure and coordination. (**A**) Model of the entire cyclic peptide, including the residues that lack density (gray), with the Vps4 pore loop 1 and 2 residues. (**B**) Superposition of the cyclic peptide structure (colors) on the previously determined structure of the linear peptide F (gray). The returning strand of the cyclic peptide is omitted for clarity. (**C**) Ordered residues of the cyclic peptide are shown as green ribbons. Pore loop one residues K205, W206, and M207 of the five Vps4 subunits that form the helical assembly that binds the substrate peptide are shown as sticks and molecular surfaces. Alternating side chains bind to class I pockets between pairs of W206 residues of adjacent subunits (two examples labeled), and to class II pockets between pairs of M207 side chains from adjacent subunits (one example labeled).

DOI: https://doi.org/10.7554/eLife.44071.027

The following videos are available for figure 3:

**Figure 3—video 1.** Cyclic peptide complex structure.
DOI: https://doi.org/10.7554/eLife.44071.028

**Figure 3—video 2.** Superposition of the cyclic peptide complex with the previously reported linear peptide complex (gray) (*Han et al., 2017*).
DOI: https://doi.org/10.7554/eLife.44071.029

**Figure 3—video 3.** The primary strand of the cyclic peptide (dark green) binds the Vps4 pore loop one residues.
DOI: https://doi.org/10.7554/eLife.44071.030

transition is coupled to conformational changes that may propagate through an extended 10-residue strand to the subunit E finger arginine residues that complete coordination of ATP at the subunit D active site (*Figure 6*). This proposal extends the model that ATP is stably bound at the AB, BC, CD interfaces, while ATP hydrolysis is promoted at the DE interface by the ATP-dependent binding of subunit F against subunit A.

## Structure and functional implications of the Vta1 activator protein

The Vta1[VSL] domains dimerize in an antiparallel orientation through formation of a four-helix bundle (*Xiao et al., 2008*). Each dimer binds to adjacent Vps4 subunits, yielding a stoichiometry of 12 Vta1 subunits (six dimers) to 6 Vps4 subunits. The Vta1[VSL] density is clearer than our earlier reconstructions with the extended Vps2 peptide (*Han et al., 2017*; *Monroe et al., 2017*), presumably because of its 10-fold higher concentration in the current study and consequently higher occupancy in the particles imaged. Moreover, the Vta1[VSL] density is better for the cFF30 structure compared to the cF30 dataset, presumably because the use of Spotiton to prepare cFF30 grids reduced the time to vitrification, which reduced the contacts with the air-water interface that cause complex dissolution and preferred orientation (*Dandey et al., 2018*; *Noble et al., 2018*).

In a substantial increase over our previous reconstructions using conventional blotting methods (*Han et al., 2017*; *Monroe et al., 2017*), ~17% of the particles derived from Spotiton grids

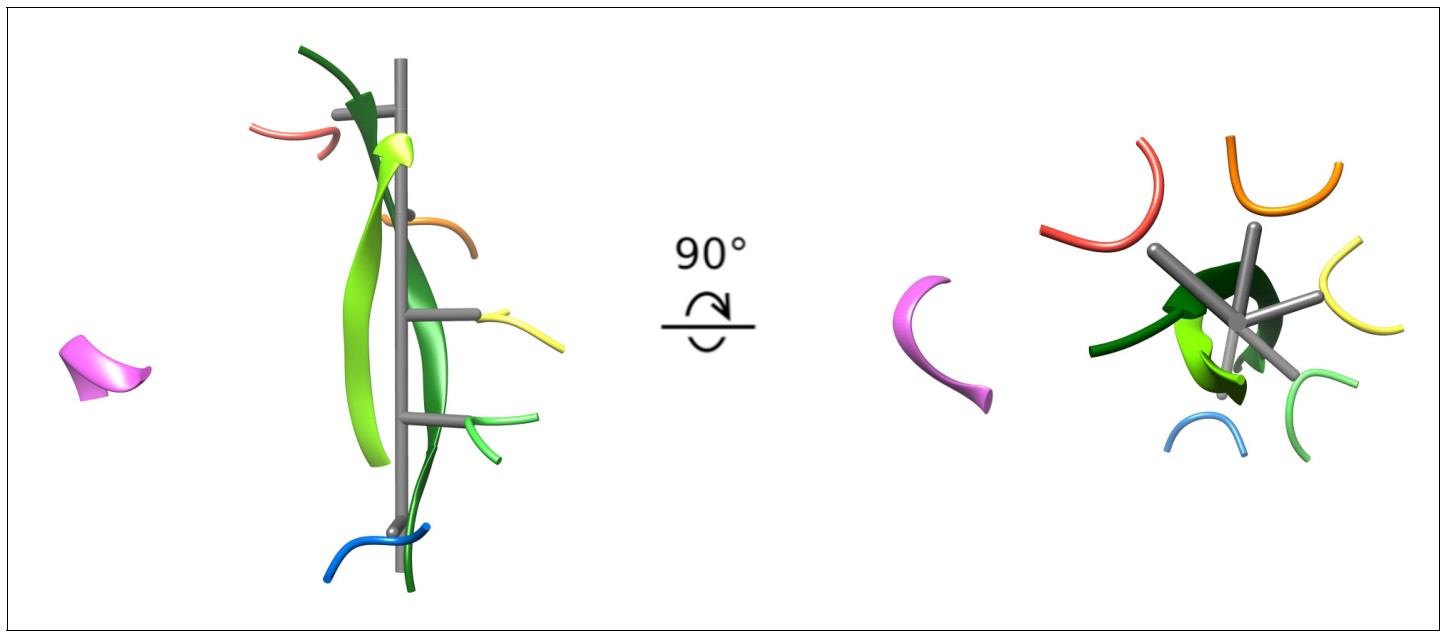

**Figure 4.** Cyclic peptide aligns with the helical axis. Side and top views of the cyclic peptide and Vps4 pore loop one residues. The pseudo two-fold axis that relates the path of the two peptide strands to each other (albeit with opposite direction) aligns with the helical axis of Vps4 subunits A-E (gray). This ensures that the second strand is maximally distant from subunits A-E, thereby explaining the lack of contacts between Vps4 and the second strand.

DOI: https://doi.org/10.7554/eLife.44071.031

The following video is available for figure 4:

**Figure 4—video 1.** The Vps4 pore is shown with peptide removed, followed by inclusion of just the primary strand (which superimposes on the previously reported linear peptide structure), and then with both strands of the cyclic peptide complex shown.

DOI: https://doi.org/10.7554/eLife.44071.032

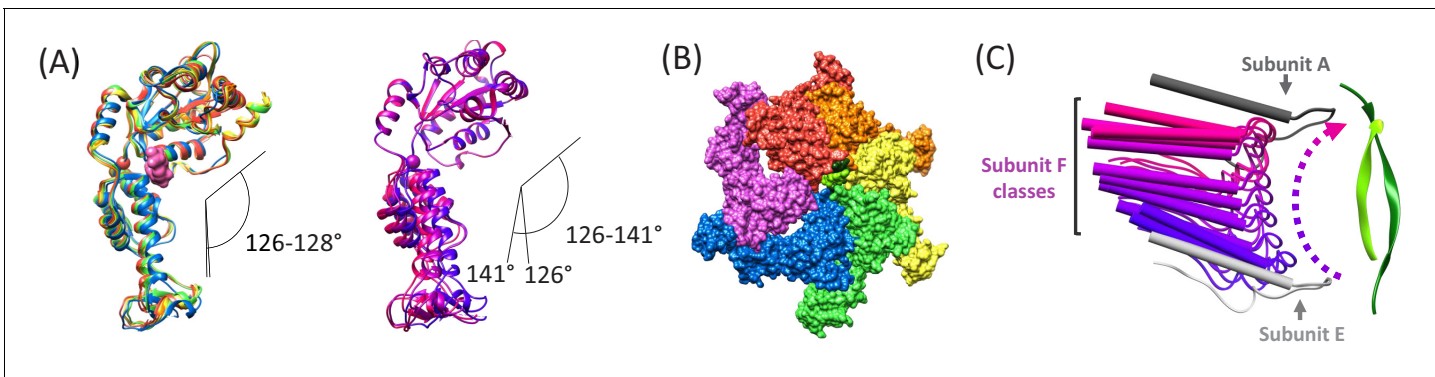

**Figure 5.** Subunit F conformation and contacts. (A) Overlap of subunits showing variation in the hinge angle between large and small domains, as defined in *Gonciarz et al. (2008)*. The overlaps were performed on the large domains. *Left,* Subunits A-E. *Right,* three classification structures of subunit F. (B) The large domain interfaces for AB, BC, CD, and DE subunit pairs are closely associated. These interfaces are much more open for the EF and FA subunit pairs, as seen by the openings in this slightly tilted view. (C) Side view of the Vps4 circular peptide complex with all of the classified subunit F models from all structures: cF30, cFF30, and combined cF30 and cFF30, plus the two previously reported data sets of Vps4 with the linear peptide F (*Han et al., 2017*; *Monroe et al., 2017*). Subunit F is colored based on position along its proposed trajectory.

DOI: https://doi.org/10.7554/eLife.44071.033

The following video is available for figure 5:

**Figure 5—video 1.** Shown as linear interpolation between the different subunit states seem in the combined cF30 and cFF30 reconstruction, including the subunit F classifications.

DOI: https://doi.org/10.7554/eLife.44071.034

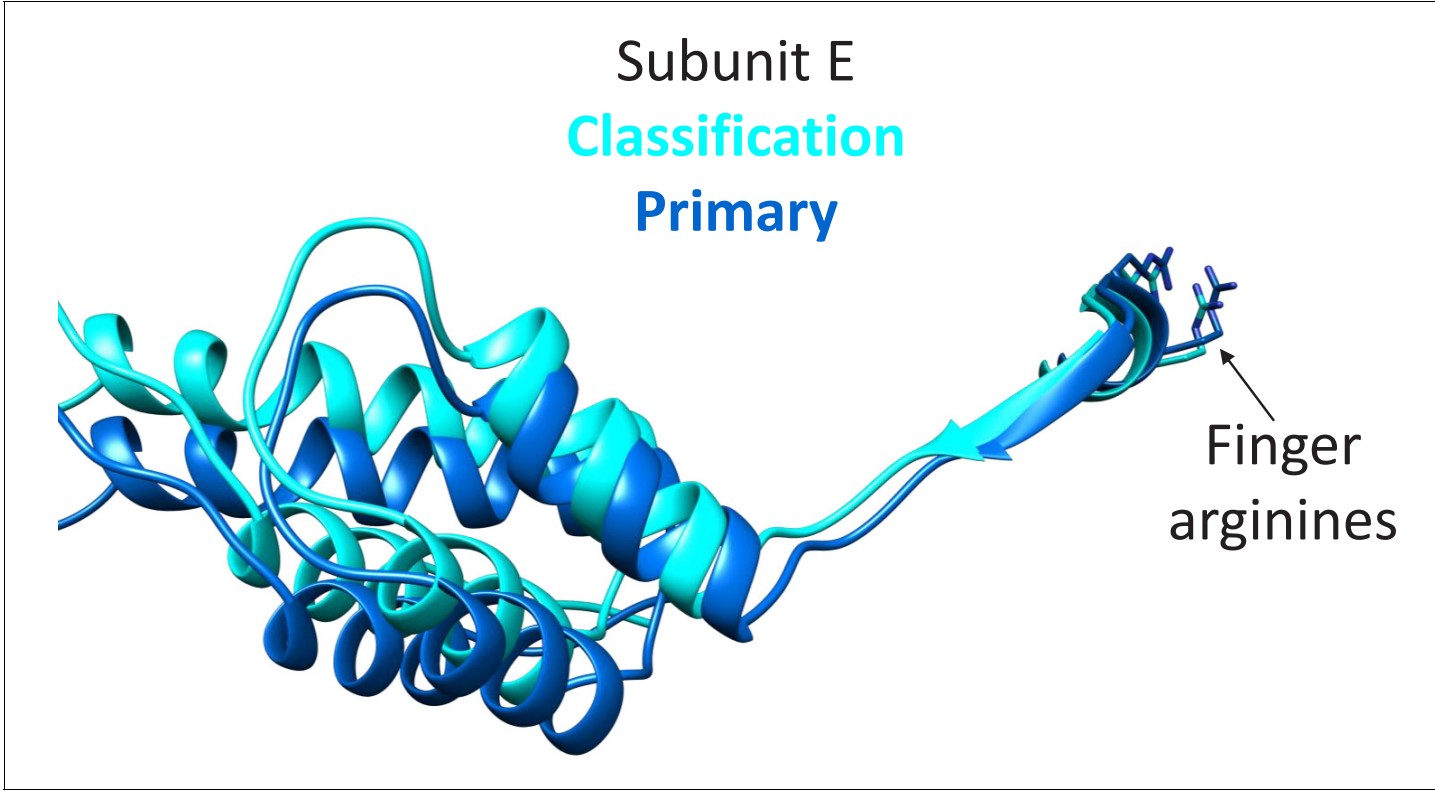

**Figure 6.** Transition of subunit F is coupled to movement of subunit E. The maps reconstructed by focused classification over subunit F show that the uppermost subunit F position from the cyclic peptide data correlates with displacement of the subunit E small domain (cyan). This domain is connected by an extended 10-residue stretch of residues to the finger arginines that complete the subunit D active site.
DOI: https://doi.org/10.7554/eLife.44071.035

contribute density to all six Vta1^VSL sites on the Vps4 hexamer (*Figure 2—figure supplement 8*). Reconstruction of these particles yielded an overall resolution of 4.4 Å. Distinct densities at each Vps4 subunit interface allowed unambiguous assignment of all four helices in the Vta1 dimers, although their location at the periphery of the Vps4 complex remains a region of relatively low local resolution (~5–8 Å). This supports the conclusion that Vps4 and Vta1 associate with 6:12 stoichiometry in the fully assembled complex (*Sun et al., 2017*).

Vta1 promotes Vps4 oligomerization and increases the ATPase activity (*Azmi et al., 2006*; *Lottridge et al., 2006*; *Scott et al., 2005*), and enhances ESCRT-III disassembly in vitro (*Azmi et al., 2008*). These effects likely result from multiple interconnected mechanisms: Association of Vta1's N-terminal MIT domains with ESCRT-III polymers will reinforce Vps4 recruitment to ESCRT-III complexes; binding of Vta1^VSL dimers in bridging interactions between adjacent Vps4 subunits will promote formation of the active hexamer; and Vta1^VSL support the contacts of subunits E and F that may promote ATP hydrolysis at the subunit D active site (above).

## Comparison with other AAA+ ATPase peptide complexes

Several structures of AAA+ ATPases that translocate protein substrates have been reported with coordinates of bound peptides deposited in the Protein Data Bank, including Hsp104 (*Gates et al., 2017*), YME1 (*Puchades et al., 2017*), TRIP13 (*Alfieri et al., 2018*), the proteasome (*de la Peña et al., 2018*; *Dong et al., 2019*), and NSF (*White et al., 2018*). In all cases the bound peptides are single, extended strands that pass through the pore and overlap closely with the structure of peptide F in the Vps4 complex, albeit sometimes in the opposite orientation or with local differences in phi/psi angles. Superposition on the pore loop 1 residues of Vps4 shows that the circular Vps4-bound cF30/cFF30 peptide fits reasonably into the YME1, Hsp104, and NSF structures (*Figure 7*, *Figure 7—video 1*). In contrast, superposition on the TRIP13 and proteasome structural models

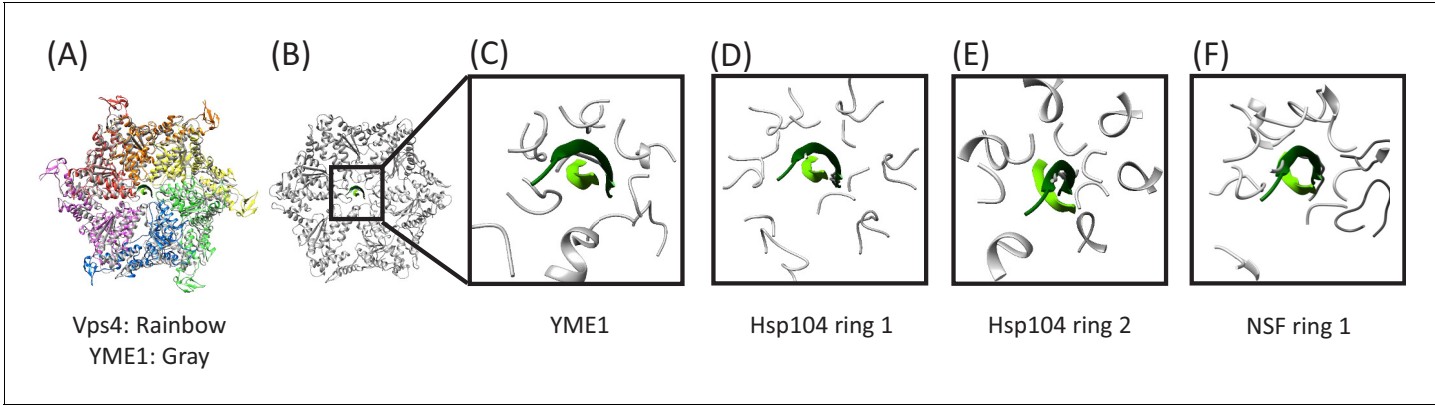

**Figure 7.** Overlap of cyclic peptide structure suggests compatibility with YME1, Hsp104, and NSF. (A) Vps4 color, YME1 (*Puchades et al., 2017*) gray. Overlap is on the Cα atoms of the peptide and pore loop 1 residues of the five helical subunits. (B) Same as (A) but without Vps4 and showing the zoomed in region of panels (C), (D), and (E). (C) Vps4 cyclic peptide after overlap on YME1. (D) Same as (C) for Hsp104 ring 1 (*Gates et al., 2017*). (E) Same as (C) for Hsp104 ring 2. (F) Same as (C) for NSF (*White et al., 2018*).

DOI: https://doi.org/10.7554/eLife.44071.036
The following videos are available for figure 7:
**Figure 7—video 1.** Overlap with other AAA+ ATPases.
DOI: https://doi.org/10.7554/eLife.44071.037
**Figure 7—video 2.** Model for AAA+ ATPase translocation of substrate from an internal loop.
DOI: https://doi.org/10.7554/eLife.44071.038
**Figure 7—video 3.** Model for AAA+ ATPase translocation of a ubiquitylated substrate.
DOI: https://doi.org/10.7554/eLife.44071.039

show overlap of pore loop two residues with the returning strand of the cyclic peptides, although in general the pore loop two residues appear to be relatively mobile, which raises the possibility that they may reposition to accommodate a two-stranded substrate. Thus, regardless of whether or not Vps4 binds its substrates in a hairpin conformation in vivo, it seems possible that this mechanism will also be accessible to other AAA+ ATPases.

## Implications for mechanism and function

Given the model that substrate translocation results from the pore loops acting on just one of the strands, it will be interesting to determine the extent to which the two chains of dual-stranded substrates slip with respect to each other, and the extent to which they pass through the pore at the same rate. Our structures also raise interesting questions about directionality. Although it is not yet definitively resolved, the Vps4 structures are most consistent with binding of the primary strand N-terminal residues at the A-end of the Vps4 helix and C-terminal residues at the E-end. This orientation is consistent with the biological role of Vps4 in translocating toward the ESCRT-III N-terminal domain, but other AAA+ ATPases apparently translocate their protein substrates in the opposite direction (N-to-C) (*Alfieri et al., 2018*; *Puchades et al., 2017*) or in either direction (*Augustyniak and Kay, 2018*). Indeed, the same mechanism of translocation could be applied to substrates bound with their primary strand in either orientation because side chains make a major contribution to binding, and forward and reversed β-strands can superimpose their Cα atoms and side chains. Regardless, our structures suggest that substrates might bind the AAA+ pore in a hairpin conformation that is translocated in both the N-to-C and C-to-N directions at the same time.

Contexts in which translocation of two polypeptide chains through a AAA+ ATPase pore might be functionally important include the initiation of translocation from an internal loop, crosslinked substrates, and ubiquitin adducts, as have been indicated for the proteasome (*Kraut and Matouschek, 2011*; *Lee et al., 2002*; *Shabek and Ciechanover, 2010*), Cdc48 (*Bodnar and Rapoport, 2017*), and ClpX (*Burton et al., 2001*). The model for translocation from an internal loop is illustrated in *Figure 7—video 2* and the model for translocation of a ubiquitylated substrate is illustrated in *Figure 7—video 3*. As shown in these animations, the model for simultaneous translocation of

two polypeptide chains through the pore of a AAA+ ATPases will cause them to rotate with respect to each other, with one complete rotation anticipated for every 12 residues of primary strand translocated. This is because successive dipeptides of the primary strand bind successive Vps4 subunits, which are rotated 60° with respect to each other, at the lining of the pore, while the secondary strand continually lies along the center of the pore. The twist that this introduces to the substrate when long stretches of double chains are translocated might be resolved by relative rotation of the AAA+ ATPase or by periodically switching which strand engages the pore loops and which lies along the open channel. The extent to which simultaneous binding and translocation of two polypeptide chains or a polypeptide with some other adduct actually occurs for the various different AAA+ ATPases remains to be determined.

# Materials and methods

## Key resources table

| Resource type | Designation | Source or reference | Identifiers |
|---|---|---|---|
| Recombinant protein | Vps4-HCP1 | PMID: 28379137 | Addgene (RRID:SCR_002037): 87737 |
| Recombinant protein | Vta1[VSL] | PMID: 28379137 | Addgene (RRID:SCR_002037): 87738 |
| Software, algorithm | GraphPad Prism | GraphPad Software, Inc, La Jolla, C | RRID:SCR_002798 |
| Software, algorithm | MotionCor2 | PMID: 28250466 | RRID:SCR_016499 |
| Software, algorithm | RELION | PMID: 23000701 | RRID:SCR_016274 |
| Software, algorithm | UCSF Chimera | PMID: 15264254 | RRID:SCR_004097 |
| Software, algorithm | Coot | PMID: 20383002 | RRID:SCR_014222 |
| Software, algorithm | Phenix | PMID: 20124702 | RRID:SCR_014224 |

## Materials used for peptide synthesis

2-chlorotrityl resin and 1-[Bis(dimethylamino)methylene]−1 H-1,2,3-triazolo[4,5-b]pyridinium-3-oxid-hexafluorophosphate (HATU) were purchased from ChemPep. TentaGel R RAM resin was purchased from Rapp Polymere. Boc-Cys(Trt)-OH, Fmoc-L-Cys(Trt)-OH, Fmoc-L-Lys(Boc)-OH, Fmoc-L-Asp(tBu)-OH, Fmoc-L-Glu(tBu)-OH, Fmoc-L-Ser(tBu)-OH, Fmoc-L-Asn(Trt)-OH, Fmoc-Gly-OH, Fmoc-L-Ala-OH, Fmoc-L-Val-OH, Fmoc-L-Leu-OH, and Fmoc-Gly-Ser(psiMe,Mepro)-OH were purchased from Gyros Protein Technologies. Fmoc-Lys(Ac)-OH was purchased from Anaspec. Fmoc-Lys(Dde)-OH was purchased from AAPPTec. Synthesis grade trifluoracetic acid (TFA), ACS grade dimethylformamide (DMF), peptide synthesis grade n-methylmorpholine (NMM), synthesis grade n-methylpyrrolidinone (NMP), ACS grade anhydrous diethyl ether, HPLC grade acetonitrile (ACN), HPLC grade methanol, LC-MS grade ACN with 0.1% formic acid, and LC-MS grade water with 0.1% formic acid were purchased from Fisher Scientific (all reagent brands from Fisher Scientific were Fisher Chemical). Piperidine, triisopropylsilane (TIS), 1,2-ethanedithiol (EDT), 5 (6)-carboxyfluorescein, N,N′-Diisopropylcarbodiimide, Oxyma Pure, anhydrous hydrazine, acetic anhydride, and 4-mercaptophenylacetic acid (MPAA) were purchased from Sigma Aldrich.

## Peptide synthesis

Peptides were synthesized on a Prelude X instrument (Gyros Protein Technologies) using Fmoc solid-phase peptide synthesis at 30 μmol scale. Deprotection cycles employed three treatments of 2 mL 20% piperidine in DMF for 3 min followed by three washes for 30 s using 2 mL DMF. Coupling cycles consisted of addition of 0.65 mL 200 mM amino acid in NMP, 0.65 mL 195 mM HATU in DMF, and 0.5 mL 600 mM NMM in DMF. Resin and coupling reagents were then mixed using nitrogen for

25 min at room temperature before being washed three times with 2 mL DMF. Tentagel R RAM resin (loading density 0.19 mmol/g) was utilized for the synthesis of C-terminal amides on peptides F, F12, F30, and FF30. To generate C-terminal hydrazides for cF30 and cFF30, 2-chlorotrityl chloride resin was converted to 2-chlorotrityl hydrazine at 0.2 mmol/g density according to published protocol (*Zheng et al., 2013*). To improve synthesis quality of peptides, the pseudoproline Fmoc-Gly-Ser ($\Psi^{Me,Me}$pro)-OH was introduced. Labeled peptide F was generated through the coupling of 5-(6)-carboxyfluorescein at the N-terminus. After completion of syntheses, peptide resins were thoroughly washed with DCM and dried under vacuum. Cleavage of peptide resins was achieved after 180 min agitation with 4 mL TFA containing 2.5% each of water, TIS, and EDT per 30 μmol peptide resin. The TFA solution was then precipitated into ice-cold diethyl ether and centrifuged at 4696 g (5,000 RPM) for 10 min. Supernatant was decanted while pellets were triturated with ether before being dried under vacuum.

## Peptide cyclization by native chemical ligation

The approach to peptide cyclization was adapted from previous work (*Zheng et al., 2012*). 1.5 μmol of HPLC purified linear peptide with C-terminal hydrazide was dissolved in 9 mL deionized (DI) water and stored on ice for 30 min. 1 mL of 200 mM NaNO$_2$ (pH 3.75 in DI water) was then added and allowed to react for 20 min at 4°C to convert the hydrazide into an acyl azide. Conversion of the acyl azide into a 4-mercaptophenylacetic acid (MPAA) thioester was achieved through addition of 10 mL 100 mM MPAA in ligation buffer (6 M GnHCl, 200 mM PO$_4$, pH 7.2). The reaction was then nutated at room temperature. At 15 min, 2 mL (1:1 ligation buffer, DI water, pH 7) 0.5 M TCEP was added to reduce oxidized MPAA and peptide. The reaction was quenched after 1 h with 1 mL 100% AcOH before centrifugation at 4,696 g for 15 min. This solution was filtered with a 0.2 μm syringe filter before HPLC purification.

## Peptide purification and analysis by HPLC and LC-MS

30 μmol of crude peptide was dissolved in 20% ACN 0.1% TFA and sonicated for 5 min before centrifugation at 4696 g for 10 min to remove precipitated material. Supernatant was filtered using a 0.2 μm filter before injection onto HPLC. Mobile phases for purification were 0.1% TFA in water (Buffer A) and 0.1% TFA in 90% ACN (Buffer B). Purification was performed on an Agilent 1260 or Beckman Gold 126 HPLC while analytical traces were collected on an Agilent 1260 HPLC. Conditions for each peptide purification are detailed in *Table 2*. Analytical traces were collected at 214 nm over a 20 min gradient of 10% to 55% Buffer B at 1 mL/min using a Phenomenex C18 Kinetix column (100 Å, 5 μm, 4.6 × 150 mm) heated to 45°C, except where noted within figure supplement legends (*Figure 1—figure supplements 2A*, *3A* and *4A*, *5A/C*, *6A*, and *7A/C*).

For LC-MS analysis, mobile phases were 0.1% FA in water (Buffer A) and 0.1% FA in ACN (Buffer B). Data were collected on an Agilent 6120 single quadrupole mass spectrometer with an Agilent 1260 front-end. HPLC traces were collected over a 7 min gradient of 5% to 90% Buffer B at 0.75 mL/min using an Agilent Poroshell C18 column (120 Å, 3.6 μm, 4.6 × 50 mm) heated to 50°C. Mass

**Table 2.** HPLC purification conditions for the peptides used in this study.

| Peptide | HPLC column | Flow rate | Gradient |
|---|---|---|---|
| F | Phenomenex C4 Luna (100 Å, 10 μm, 10 × 250 mm) | 5 mL/min | 20 to 80% Buffer B |
| F12 | Phenomenex C12 Jupiter (90 Å, 10 μm, 21.2 × 250 mm) | 5 mL/min | 25 to 45% Buffer B |
| F30 | Phenomenex C12 Jupiter (90 Å, 10 μm, 21.2 × 250 mm) | 10 mL/min | 20 to 45% Buffer B |
| cF30 (pre-cyclization) | Phenomenex C12 Jupiter (90 Å, 10 μm, 21.2 × 250 mm) | 10 mL/min | 20 to 45% Buffer B |
| cF30 | Phenomenex C4 Jupiter (100 Å, 10 μm, 10 × 250 mm) | 5 mL/min | 10 to 35% Buffer B |
| FF30 | Phenomenex C12 Jupiter (90 Å, 10 μm, 21.2 × 250 mm) | 10 mL/min | 29 to 37% Buffer B |
| cFF30 (pre-cyclization | Phenomenex C12 Jupiter (90 Å, 10 μm, 21.2 × 250 mm) | 10 mL/min | 20 to 45% Buffer B |
| cFF30 | Phenomenex C4 Jupiter (100 Å, 10 μm, 10 × 250 mm) | 5 mL/min | 27 to 38% Buffer B |

DOI: https://doi.org/10.7554/eLife.44071.040

spectra were obtained over a window of 400 to 2,000 *m/z* in fast scan and positive ion mode (*Figure 1—figure supplements 2B*, *3B* and *4B*, *5B/D*, *6B* and *7B/D*). Deconvoluted masses were determined using Agilent Chemstation with averaged scans across the major ion signal.

## Trypsin digestion of peptides and high-resolution LC-MS

Linear and cyclic peptides from lyophilized powder were dissolved in alkylating buffer (40 mM 2-chloroacetamide, 10 mM TCEP, 100 mM Tris, pH 8) to a concentration of ~30 µM before rotating at 37°C for 60 min. Before sample loading, Pierce C18 tips were equilibrated with three treatments of 100 µL 0.1% FA in ACN and three treatments of 100 µL 0.1% FA in water. 40 µL of each sample loaded onto Pierce C18 tips through five repeats of aspiration and dispensing. C18 tips were washed with 100 µL 0.1% FA in water before elution of the peptide using 0.1% FA in 70% ACN. The eluent was concentrated to 10 µL by speed-vac before addition of 50 µL trypsin (1:10, Pierce Trypsin Protease MS Grade, Thermo Fisher Scientific) in 50 mM ammonium bicarbonate buffer (ABC, pH 7.5). Trypsin treated samples were incubated at 37°C for 90 min before quenching through addition of FA to a final concentration of 1%. Trypsin was removed from the samples using Vivacon 30 k MWCO filters (Sartorius) and centrifugation at 14,000 x g for 20 min. Samples were then diluted 1:1000 using 0.1% FA in water before transferring to MS vials and storage at −80°C.

For mass spectrometry analysis, 2 µL of sample was injected onto a Thermo Fisher EASY-nanoLC 1000 with a Picofrit column (New Objectives, 360 µm OD x 75 µm ID, 150 mm, packed with 3 µm Reprosil-PUR) coupled to an Orbitrap Velos Pro. Mobile phases consisted of 0.1% FA 5% DMSO in water (Buffer A) and 0.1% FA 5% DMSO in ACN (Buffer B). Gradient conditions were 5% to 45% Buffer B over 30 min at 400 nL/min. MS1 spectra were collected using the Orbitrap analyzer from 350 to 1550 *m/z* at 60,000 resolution (FWHM as defined at *m/z* 400). The top two most intense ions from the MS1 scan were selected for HCD fragmentation using a normalized collision energy of 40 eV. MS2 spectra were collected at a resolution of 15,000.

To identify peptides in an unbiased manner, raw data files were converted to mgf format for analysis in SearchGUI (*Barsnes and Vaudel, 2018*) using the OMSSA search algorithm. Peptide sequences were added to a modified FASTA file containing ~2000 decoy human proteins. Spectrum match settings used trypsin digestion with one missed cleavage allowed, carbamidomethylation of Cys as a fixed modification (denoted as 'Am' in peptide sequences), acetylation of K as a variable modification, precursor *m/z* tolerance of 10 ppm, and fragment *m/z* tolerance of 0.2 Da. Post-processing utilized PeptideShaker (*Vaudel et al., 2015*) to identify peptide fragments present in each sample. The false discovery rate (FDR) for peptide identification was set to 0.01. Identified peptides were exported from PeptideShaker as an MZID file to Skyline (*MacLean et al., 2010*). Visualization of all peptide fragments (*Figure 1—figure supplement 8*) and MS2 spectra (*Figure 1—figure supplement 9*) were produced using Skyline.

## Fluorescence polarization assay for peptide binding

Binding of unlabeled peptides to Vps4 and Vps4-Hcp1 hexamer was quantified using competitive binding assays. Briefly, a dilution series of unlabeled peptide was made in fluorescence polarization assay buffer (20 mM HEPES, pH 7.4, 100 mM NaCl, 1 mM ADP·BeF$_x$, 10 mM MgCl$_2$, 10 mM TCEP) with 200 nM Vps4 hexamer and 1 nM fluorescein-labeled peptide F. Reaction equilibrium was reached with 3 h incubation at room temperature. Fluorescence polarization was measured on a Biotek Synergy Neo HTS Multi-Mode Microplate Reader using 485/528 nm excitation/emission wavelengths. IC$_{50}$ values were calculated using GraphPad Prism 7 (RRID:SCR_002798) by fitting raw polarization data to *Equation 1* with the FP$_{min}$ manually constrained to the polarization value of labeled peptide F alone. $K_i$ values, which correspond to the dissociation constants for the unlabeled peptides, were calculated with *Equation 2* (*Nikolovska-Coleska et al., 2004*) using previously published $K_D$ values for labeled peptide F (0.253 ± 0.015 µM with Vps4 and 0.230 ± 0.010 µM with Vps4-Hcp1; *Monroe et al., 2017*) and IC$_{50}$ values from *Equation 1*.

$$FP = FP_{min} + \frac{FP_{max}FP_{min}}{1 + \left(\frac{[unlabeled\ peptide]}{IC_{50}}\right)} \tag{1}$$

$$K_i = \frac{[I]_{50}}{\frac{[L]_{50}}{K_D} + \frac{[P]_0}{K_D} + 1}$$

(2)

## Grid preparation and vitrification

Complexes were prepared for cryo-EM analysis as described (*Monroe et al., 2017*) except that Vta1$^{VSL}$ was included at 10-fold higher concentration for crosslinking. cF30 complex was vitrified using a Vitrobot (Thermo Fisher Scientific), as described (*Monroe et al., 2017*).

cFF30 complex samples were vitrified using the Spotiton robot as described (*Dandey et al., 2018*; *Jain et al., 2012*; *Razinkov et al., 2016*), starting from a Vps4 complex at 18 µM (hexamer). Briefly, the Spotiton device uses piezo dispensing to apply small (50 pL) drops of sample across a 'self-blotting' nanowire grid as it flies past *en route* to plunge into liquid ethane. Nanowire grids for use with Spotiton were manufactured in-house, backed by lacey carbon film supports, and prepared as described (*Razinkov et al., 2016*; *Wei et al., 2018*), including plasma cleaning for 10 s (O$_2$ + H$_2$) using a Solarus 950 (Gatan, Inc). The time between sample application to the grid and plunging into liquid ethane (spot-to-plunge time) was ~145 ms. Spotiton was operated at ~85% relative humidity and ambient temperature (~21°C). Under these conditions, evaporation is estimated to be 300 Å/s.

## Single-particle cryoEM data collection

Single-particle micrographs were collected on a Titan Krios (Thermo Fisher Scientific) equipped with an energy filter and a K2 BioQuantum counting camera (Gatan, Inc); the microscope was operated at 300 kV at a nominal magnification of 130,000x, with a calibrated pixel size of 1.09 Å. Exposure was set to 10 s (50 frames/movie, detector operated at counting mode), for a total dose of 76.68 e$^-$/Å$^2$ with a defocus range of 1.6 to 2.2 µm. Each dataset was collected over one session using Legion. Frames were aligned and dose weighted using MotionCor2 RRID:SCR_016499 (*Zheng et al., 2017*).

## Image processing

cF30 and cFF30 complex datasets were initially processed separately with essentially the same workflow (*Figure 2—figure supplements 2* and *3*). Movie frames were aligned, dose weighted, and summed using MotionCor2 RRID:SCR_016499 (*Zheng et al., 2017*). CTF parameters were determined on non-dose-weighted sums using gctf (*Zhang, 2016*). Micrographs with poor CTF cross correlation scores (<0.04) were excluded from downstream analyses. A total of 2838 (cF30) and 1855 (cFF30) dose-weighted sums were used for all subsequent image processing steps.

529,807 (cF30) and 356,705 (cFF30) particles were extracted and used as input for full CTF-corrected image processing. After multiple rounds of 2D classification in RELION (RRID:SCR_016274) (*Scheres, 2012*), 204,636 (cF30) and 155,900 (cFF30) particles were retained, based on visual inspection of classes with high-resolution Vps4 features, and used for an initial round of 3D classification. After 3D classification, 157,775 (cF30) and 144,776 (cFF30) particles were used for RELION (RRID:SCR_016274) auto-refinement (*Scheres, 2012*), which in each case generated an ~4.5 Å density map of the Hcp1-Vps4 fusion complex based on the gold-standard FSC criterion. To improve the resolution of Vps4, we performed signal subtraction of Hcp1 densities, as described (*Bai et al., 2015*; *Monroe et al., 2017*). For the cF30 complex dataset, an additional round of 3D classification post-Hcp1-subtraction assigned 92,704 particles into a single class with high-resolution Vps4 features. These particles were used for a final round of RELION (RRID:SCR_016274) auto-refinement, producing a 3.8 Å resolution density map of Vps4. For the cFF30 complex dataset, the Hcp1-subtracted dataset was used for RELION (RRID:SCR_016274) auto-refinement without further classification, producing a 4.0 Å resolution density map.

To improve the resolution of the density map, the final datasets for cF30 and cFF30 complexes were combined for further processing (*Figure 2—figure supplements 1* and *4*). Merging the datasets was justified based on the similar binding affinities and densities (*Figure 2D*). The merged dataset comprised a total of 237,480 particles (92,704 from cF30 complex, 144,776 from the cFF30 complex). RELION (RRID:SCR_016274) auto-refinement was performed using the –solvent_correct_fsc option in combination with a soft-edge mask encompassing Vps4. B-factor sharpening of −157 Å$^2$ was applied to the final 3.56 Å map using an automated procedure in RELION (RRID:SCR_

016274) postprocessing (*Rosenthal and Henderson, 2003*). Local resolutions were estimated using ResMap (*Kucukelbir et al., 2014*).

As observed in our previous structures of Vps4 bound to linear peptides, subunit F was poorly resolved due to its relative flexibility (*Monroe et al., 2017*). We therefore performed focused 3D classification by applying a spherical mask over subunit F density (*Figure 2—figure supplement 4*). Classification was performed without re-alignment (i.e., using the –skip_align flag in RELION, K = 6), leading to three classes with ordered density and three classes with disordered density (*Figure 2—figure supplement 5*). Particles from classes with ordered density were used for separate RELION auto-refinement. The resulting maps were then used for rigid-body fitting of Vps4 coordinates into each subunit F position using UCSF Chimera (RRID:SCR_004097) (*Pettersen et al., 2004*).

Focused 3D classification of Vta1$^{VSL}$ was performed by applying a single custom mask that covered each of the six possible Vta1 binding sites (*Figure 2—figure supplement 8*). Particles were classified without re-alignment (i.e., using the –skip_align flag in RELION, K = 10). 24,778 particles from the cFF30 dataset were sorted into a single class that showed Vta1$^{VSL}$ densities at all six Vps4 interfaces. These particles were used for RELION 3D auto-refinement, resulting in a 4.4 Å resolution reconstruction. Vta1$^{VSL}$ models were fitted into the reconstruction using rigid-body fitting as previously described (*Monroe et al., 2017*).

## Model building, refinement, and validation

The model of AAA+ ATPase cassettes for Vps4 subunits A-E and the substrate from our previous structure (PDBID: 6AP1) were fit to the 3.6 Å map as rigid bodies and subjected to real-space refinement using Phenix (RRID:SCR_014224) (*Adams et al., 2010*) following the same approach as for the earlier structure of Vps4 in complex with a linear substrate (*Han et al., 2017*). The returning chain was built manually in Coot (RRID:SCR_014222) (*Emsley et al., 2010*) followed by real-space refinement using Phenix with restraints to β-strand conformation and its starting position. The refined model was assessed using MolProbity (RRID: SCR_014226) (*Chen et al., 2010*).

To test for overfitting, the refined model (subunits A-E of Vps4 and the main and returning chains of the cyclic substrate) were randomly displaced by 0.2 Å and re-refined against one of the RELION half-maps used to generate the 3.6 Å map. FSC curves were generated for the re-refined model against the half map used for re-refinement (FSC$_{work}$) and against the other half map (FSC$_{test}$) (*Figure 2—figure supplement 1E*). The agreement between FSC$_{work}$ and FSC$_{test}$ indicates that the model has not been overfit.

## Structure deposition

The refined model comprising the Vps4 ATPase domains of subunits A-E and the cyclic peptide is accessible via the PDB (RRID: SCR_012820; PDB ID: 6NDY) together with the 3.6 Å map from the combined dataset (RRID: SCR_003207, EMDB Accession Number EMD-0443). The complete model, including regions not subjected to atomic refinement such as the 12 Vta1$^{VSL}$ domains and subunit F, is also available via the PDB (PDB ID: 6OO2), together with the map containing Vta1$^{VSL}$ densities at all six Vps4 interfaces (RRID: SCR_003207, EMDB Accession Number EMD-20142). The two maps derived from the cF30 and cFF30 complex datasets individually, and the three maps for subunit F, have been deposited to the EMDB (RRID: SCR_003207, EMDB Accession Numbers EMD-20144, EMD-20147, EMD-20139, EMD-20140, EMD-20141).

## Acknowledgements

We thank Aman Makaju, Anna Bakhtina, and Dr. Sarah Franklin for mass spectrometry analysis of trypsin-digested samples. Electron microscopy was performed at the Simons Electron Microscopy Center and National Resource for Automated Molecular Microscopy located at the New York Structural Biology Center.

## Additional information

### Competing interests
Wesley I Sundquist: Reviewing editor, *eLife*. The other authors declare that no competing interests exist.

### Funding

| Funder | Grant reference number | Author |
|---|---|---|
| National Institutes of Health | P50 GM082545 | Han Han<br>James M Fulcher<br>Janet H Iwasa<br>Michael S Kay<br>Peter S Shen<br>Christopher P Hill |
| National Institutes of Health | R01 GM112080 | Wesley I Sundquist |
| National Institutes of Health | P41 GM103310 | Venkata P Dandey |
| New York State Foundation for Science, Technology and Innovation | | Venkata P Dandey |
| Simons Foundation | SF349247 | Venkata P Dandey |

The funders had no role in study design, data collection and interpretation, or the decision to submit the work for publication.

### Author contributions
Han Han, Formal analysis, Investigation, Visualization, Writing—review and editing; James M Fulcher, Formal analysis, Validation, Investigation, Visualization, Writing—review and editing; Venkata P Dandey, Formal analysis, Validation, Investigation, Visualization; Janet H Iwasa, Resources, Supervision, Funding acquisition, Investigation, Visualization, Project administration, Writing—review and editing; Wesley I Sundquist, Resources, Supervision, Funding acquisition, Project administration, Writing—review and editing; Michael S Kay, Resources, Data curation, Formal analysis, Supervision, Funding acquisition, Validation, Investigation, Visualization, Writing—review and editing; Peter S Shen, Conceptualization, Resources, Data curation, Formal analysis, Supervision, Funding acquisition, Validation, Investigation, Visualization, Writing—original draft, Writing—review and editing; Christopher P Hill, Conceptualization, Resources, Supervision, Funding acquisition, Validation, Investigation, Visualization, Writing—original draft, Writing—review and editing

### Author ORCIDs
Han Han (ID) https://orcid.org/0000-0003-0361-4254
James M Fulcher (ID) http://orcid.org/0000-0001-9033-3623
Janet H Iwasa (ID) https://orcid.org/0000-0002-4949-7607
Wesley I Sundquist (ID) http://orcid.org/0000-0001-9988-6021
Michael S Kay (ID) https://orcid.org/0000-0003-3186-9684
Peter S Shen (ID) http://orcid.org/0000-0002-6256-6910
Christopher P Hill (ID) https://orcid.org/0000-0001-6796-7740

### Decision letter and Author response
Decision letter https://doi.org/10.7554/eLife.44071.060
Author response https://doi.org/10.7554/eLife.44071.061

## Additional files

### Supplementary files
• Transparent reporting form

DOI: https://doi.org/10.7554/eLife.44071.041

## Data availability

The refined model comprising the Vps4 ATPase domains of subunits A-E and the cyclic peptide is accessible via the PDB (RRID: SCR_012820; PDB ID: 6NDY) together with the 3.6 Å map from the combined dataset (RRID: SCR_003207, EMDB Accession Number EMD-0443). The complete model, including regions not subjected to atomic refinement such as the 12 Vta1$^{VSL}$ domains and subunit F, is also available via the PDB (PDB ID: 6OO2), together with the map containing Vta1$^{VSL}$ densities at all six Vps4 interfaces (RRID: SCR_003207, EMDB Accession Number EMD-20142). The two maps derived from the cF30 and cFF30 complex datasets individually, and the three maps for subunit F, have been deposited to the EMDB (RRID: SCR_ 003207, EMDB Accession Numbers EMD-20144, EMD-20147, EMD-20139, EMD-20140, EMD-20141).

The following datasets were generated:

| Author(s) | Year | Dataset title | Dataset URL | Database and Identifier |
|---|---|---|---|---|
| Han H, Fulcher JM, Dandey VP, Sundquist WI, Kay MS, Shen P, Hill CP | 2019 | Vps4 with Cyclic Peptide Bound in the Central Pore | http://www.rcsb.org/structure/6NDY | Protein Data Bank, 6NDY |
| Han H, Fulcher JM, Dandey VP, Sundquist WI, Kay MS, Shen P, Hill CP | 2019 | Vps4 with Cyclic Peptide Bound in the Central Pore | http://www.ebi.ac.uk/pdbe/entry/emdb/EMD-0443 | EMDataBank, EMD-0443 |
| Han H, Fulcher JM, Dandey VP, Sundquist WI, Kay MS, Shen P, Hill CP | 2019 | Vps4 with Cyclic Peptide Bound in the Central Pore | http://www.rcsb.org/structure/6OO2 | Protein Data Bank, 6OO2 |
| Han H, Fulcher JM, Dandey VP, Sundquist WI, Kay MS, Shen P, Hill CP | 2019 | Vps4 with Cyclic Peptide Bound in the Central Pore | http://www.ebi.ac.uk/pdbe/entry/emdb/EMD-20142 | EMDataBank, EMD-20142 |
| Han H, Fulcher JM, Dandey VP, Sundquist WI, Kay MS, Shen P, Hill CP | 2019 | Vps4 with Cyclic Peptide Bound in the Central Pore (Map for Peptide cF30) | http://www.ebi.ac.uk/pdbe/entry/emdb/EMD-20144 | EMDataBank, EMD-20144 |
| Han H, Fulcher JM, Dandey VP, Sundquist WI, Kay MS, Shen P, Hill CP | 2019 | Vps4 with Cyclic Peptide Bound in the Central Pore (Map for Peptide cFF30) | http://www.ebi.ac.uk/pdbe/entry/emdb/EMD-20147 | EMDataBank, EMD-20147 |
| Han H, Fulcher JM, Dandey VP, Sundquist WI, Kay MS, Shen P, Hill CP | 2019 | Vps4 with Cyclic Peptide Bound in the Central Pore (Focused Classification of Subunit F, State1) | http://www.ebi.ac.uk/pdbe/entry/emdb/EMD-20139 | EMDataBank, EMD-20139 |
| Han H, Fulcher JM, Dandey VP, Sundquist WI, Kay MS, Shen P, Hill CP | 2019 | Vps4 with Cyclic Peptide Bound in the Central Pore (Focused Classification of Subunit F, State2) | http://www.ebi.ac.uk/pdbe/entry/emdb/EMD-20140 | EMDataBank, EMD-20140 |
| Han H, Fulcher JM, Dandey VP, Sundquist WI, Kay MS, Shen P, Hill CP | 2019 | Vps4 with Cyclic Peptide Bound in the Central Pore (Focused Classification of Subunit F, State3) | http://www.ebi.ac.uk/pdbe/entry/emdb/EMD-20141 | EMDataBank, EMD-20141 |

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
