## [Decision Letter]

[Editors’ note: this article was originally rejected after discussions between the reviewers, but the authors were invited to resubmit after an appeal against the decision.]

Thank you for submitting your work entitled "Implications for AAA ATPase processing of protein substrate loops from the structure of Vps4 bound to a circular peptide" for consideration by *eLife*. Your article has been reviewed a Senior Editor, a Reviewing Editor, and two reviewers. The reviewers have opted to remain anonymous.

Our decision has been reached after consultation between the reviewers. Based on these discussions and the individual reviews below, we very much regret to inform you that we cannot accommodate this work in *eLife*. There were no issues with the technical aspects of the findings. However, both referees were in agreement that the presented structure likely represents a specialized scenario (an initiation complex that is determined by the tightly binding F-peptide), and felt that it constitutes a variation of prior published peptide-bound structures of Vps4 which does not provide additional insights into the translocation process. There was also concern that the low complexity of the secondary strand may allow its return through the central pore without making any motor contacts, and that this may not therefore reflect a normal two-strand operation mode of AAA peptide translocases on more complex sequences. During the consultation process, the referees did note that if new conformations or substrate interactions could be identified which expand on the motor mechanism, or if an argument could be made that this the state seen here is perhaps equivalent to how Vps4 acts on Vps2 during disassembly of ESCRT-III (accompanied by appropriate functional evidence), then the impact could strengthened. We hope that you will find their comments useful in moving forward.

Reviewer #1:

In this manuscript, Hill and coworkers describe the cryo-EM structure of cyclic peptide-bound Vps4 in complex with ADP·BeF_x_ and the VSL domain of Vta1, suggesting how this AAA+ motor may translocate loops or two strands of protein substrates. The authors used circularized 30-mer peptides with one or two copies of the 8-residue segment DEIVNKVL (F peptide) that was derived from the ESCRT-II subunit Vps2, specifically binds to the Vps4 pore, and whose structure bound to Vps4 has previously been solved by the same group. This primary segment of the circular peptides was found to bind indistinguishably from the linear peptide, whereas the secondary segment forms β-ladder interactions with the primary segment and passes through the Vps4 pore without making any significant contacts.

The presented data are of good quality and provide insights into how Vsp4 may simultaneously translocate two polypeptide chains. However, the strong agreement with the previously published structure of linear peptide-bound Vps4 limits the extent of conceptually advances. Due to the use of the specifically-binding F-peptide combined with a low-complexity (G, A, and S-rich) secondary segment, it is not too surprising that affinities and peptide-motor interactions are highly similar or identical to those previously described. The authors strongly generalize their findings and suggest throughout the manuscript, including in the title and abstract, that other AAA+ ATPases may process substrate loops or multiple chains in a very similar manner. However, it is unclear to what extent the observed conformations and interactions are relevant for other AAA+ motors, as the presented structure likely reflects an initiation complex in which a specifically recognized peptide binds with micro-molar affinity to a static, ATP-hydrolysis-deficient Vps4. As acknowledged by the authors, other substrate-bound AAA+ motors do not show peptides in the same β-strand conformation with highly ordered subunit contacts, which also makes it unlikely that the secondary strand of a substrate loop forms similar β-ladder interactions as observed here for Vps4. Many of these other structures, including the recently published substrate-bound 26S proteasome, suggest less regular, steric interactions of pore-loops with the substrate polypeptide, which may then also involve contacts with a secondary strand when substrate loops are translocated.

The authors favor a model in which the primary strand binds in C-to-N terminal orientation to the Vps4 subunits A – E, with the secondary strand spared from any interactions. This model would imply that folded domains N-terminal of the motor-bound primary strand would get preferentially unfolded. Biochemical studies of substrate processing by the 26S proteasome (e.g. Piwko and Jentsch, 2006) indicate that initiation on an internal loop can lead to processing of the N-terminal, C-terminal, or both segments, speaking against selective interactions with only the primary strand. It is thus questionable whether the model presented here applies to other AAA+ motors including the proteasome.

The authors suggest (or imply through their wording) that they identified a new mechanisms of substrate engagement: "Our data indicate a third potential mechanism in which substrate engagement and translocation initiate from an internal segment by binding of a folded hairpin directly within the hexamer pore." Even though structural data have so far been missing, internal initiation and the translocation of multiple chains had already been well established by extensive biochemical studies, not only for the 26S proteasome, but also for members of the Clp family. These previous studies should be cited and the presented Vps4 structure discussed accordingly.

Regarding the coupling of ATP hydrolysis and substrate translocation, a similar coordination of conformational changes as described here for subunits F and E has also been observed for subunits of the substrate-engaged 26S proteasome in various stages of the ATPase cycle (de la Pena et al., 2018) and should be discussed.

The presented structures add some details, but no groundbreaking new findings about the interaction of Vps4 with the VSL domain of the Vta1 activator.

In summary, even though it is interesting to visualize Vps4 bound to a circular peptide, conceptually new findings are somewhat limited due to the strong similarities to the previously published structures of the linear peptide-bound motor. Since the presented structures likely represent an initiation complex with a tightly binding peptide, it remains unclear to what extent observed interactions apply to processively translocating Vps4 or even other AAA+ motors, especially in light of existing substrate-bound structures that already show deviations in peptide conformations and orientations within the central pore. The authors should try to focus their manuscript primarily on Vps4 and reduce their generalization about the mechanisms of other AAA+ motors. Contrary to the author's suggestion, these structures did not identify a novel mechanism for internal initiation of substrate processing, however, they do represent the first visualization of such a complex. I'll have to leave it to the reviewing editor to decide whether the presented advances are significant enough to consider publication in *eLife*.

Reviewer #2:

In this manuscript Han, et al. determined a 3.6-Å structure of the Vps4 AAA+ ATPase, which is essential for dissociating ESCRT complexes, bound to a cyclic peptide containing the Vps2 binding sequence. The significant results are that the peptide is bound in a β-ladder hairpin conformation with two strands spanning the Vps4 translocation channel. The Vps2 sequence is bound in the same arrangement with defined pore-loop contacts as previous structures while the returning strand is more flexible and runs along the helical interface without making contact with the hexamer. The structure is important because it shows that two strands can be accommodated in the translocation channel; all previous structures of AAA+ translocases show a single unfolded strand in the channel. The authors model this β-hairpin into these previous structures and propose this as evidence that related AAA+s can initiate and translocate internal loops of substrates in addition accessible termini – although this is tenuous given that no experiments directly address this functionally and other AAA+s are not tested. Additionally, the authors are able to increase the occupancy of the Vta1 cofactor subunit and thereby establish its stoichiometry in the complex and improve the reconstruction and modeling of this region compared to previous structures. The work in this manuscript is technically very sound with compelling substrate binding analyses, cryo-EM structure determination and molecular modeling methods. However, two points of concern regarding the functional significance and novelty of these results reduce enthusiasm for publication in *eLife*: (1) These results may be a specific consequence of the experimental setup which includes the use of a cyclic peptide with low-complexity Gly, Ala and Ser residues outside Vps2 that may be required to fit in the channel, use of ADP·BeF_x_, glutaraldehyde crosslinking, and a truncated Vps4 with the VSL domain added for stability. Thus, it is unclear whether a native Vps4 complex could bind the full-length Vps2 substrate by this mechanism or if other AAA+s can translocate multiple strands through the channel. (2) The Vps4 hexamer and bound Vps2 peptide are in an identical configuration and nucleotide state as the two previously published structures (one at higher resolution) by this group and the second strand of the substrate is flexible and passively bound without any specific contacts with the Vps4 channel, thus the insight into the AAA+ translocation mechanism, that Vps4 can accommodate two strands under these conditions, has modest impact to the field.

Specific comments:

-Were other nucleotides tested for binding to the cyclic peptide? Hydrolysable nucleotides, such as ATP or ATPyS, would be worth testing to potentially capture different nucleotide states of the subunits or different translocation intermediates.

-What is the solution structure of the cyclic peptide? Does it form the β-hairpin structure or is this a consequence of Vps4 binding? Perhaps CD spectra of the peptide could be measured.

-Were other cyclic peptides tested with difference sequences for the hairpin? It seems that this two-stranded β-ladder complex may require low complexity or specific amino acids with minimal side chains for the returning strand that is adjacent the Vps2 sequence in order to fit in the channel. Functional significance would be improved if a solution-state β-sheet or the full α-helix that contains the Vps2 sequence were tested.

-The occupancy and nucleotide state of the subunits is discussed, but no data is shown in support of this.

-In their translocation model the authors propose that subunit F, which is disconnected from the substrate and asymmetric with respect to the helical arrangement of the hexamer, moves ~30 Å to the subunit-A end of the helix during a translocation step. By focussed classification of this region of the hexamer they identify different positions of subunit F. Local resolution for this region is stated to be 4-7 Å for these classes, however no data is shown. How was the focussed classification performed? Is the cryo-EM density for F improved by focussed classification? Only the Vta1 focussed classification is discussed in the methods. From Figure 4B it is difficult to tell how these positions/conformations of F fit with their translocation model or that "they span a substantial fraction of the path" that is proposed for a translocation step. Please show the range of motion and relative positions of subunit A and E along the channel axis.

-As rationale for a potential conserved function of AAA+s binding a two-strand β-ladder structure the authors make claims that are questionable and not referenced. It is stated "AAA+ ATPases are often thought to initiate substrate engagement from a protein terminus". Please provide supporting studies for this claim. For AAA+ disaggregases it has been proposed for a number of years that engagement can occur from internal segments (see Haslberger et al., 2008).

Additionally it is stated that AAA+ hexamerization could occur "around a linear portion of their substrate", but "this possibility is relevant for family members that show inherently weak hexamerization in the absence of bound substrate but is unlikely to be applicable for robust hexamers like the proteasome and Hsp104". This claim is inaccurate, and no supporting references are provided. In fact, the "lock washer" helical conformation has only been observed in the presence of substrate for these AAA+s. Crystal structures show a continuous helix, hexamerization is dynamic and dependent on nucleotide state (see DeSantis et al., 2012, Aguado et al., 2015 and Uchihashi et al., 2018) and the substrate-free cryo-EM structures of VAT, ClpB and Hsp104 show an open helical spiral that is an entirely different arrangement and likely incompatible with translocation. Thus, while hexamerization/oligomerization can occur without substrate, these complexes are highly dynamic. Therefore, hexamerization around the substrate or, more likely, passing an internal segment though the seam interface are highly plausible models for engagement of internal segments by the proteasome, Hsp104 and other AAA+s.

[Editors’ note: what now follows is the decision letter after the authors submitted for further consideration.]

Thank you for resubmitting your work entitled "Structure of Vps4 with circular peptides and implications for translocation of two polypeptide chains by AAA+ ATPases" for further consideration at *eLife*. Your revised article has been favorably evaluated by John Kuriyan (Senior Editor), a Reviewing Editor, and two reviewers. Although one referee is now in favor of publication, the other (Reviewer 2) raises a few salient points that we feel are worth considering before making a final decision. Please respond to these comments using your best judgement. Once your revised manuscript is received, we will make an editorial decision, with no further input required from the reviewers.

Reviewer #2:

I appreciate the authors' further discussion about the F-peptide conformation and its similarities to structures observed for related AAA motors. I agree that the overall arrangement and mode of peptide interaction is similar to other motors, as this is largely dictated by the very consistent helical-staircase arrangement of ATPase subunits. The question is about the high regularity of those interactions in Vps4, whether substrates in general have to adopt a defined β structure in the pore, and to what extent the observed conformation originates from the tight binding (K_D_ = 250 nM) and consequent potential energy minimum of the characterized state. A comparison with peptide conformations in other AAA motors is certainly warranted, but needs careful phrasing, especially because the intermediate resolution of those structures makes an assessment of phi/psi angles difficult. Even for the Vps4 structure presented here, the authors' strong claims about β strand conformation is oddly contrasted by their uncertainty about N-to-C or C-to-N directionality of the bound peptide.

Independent of whether the F-peptide complex is indeed an initiation complex, its high affinity is a bit surprising given the non-specific nature of interactions during substrate translocation. If peptides in general bound with similarly high affinity, as suggested by the authors, I am wondering how Vps4 can maintain a high enough selectivity, and whether the autoinhibitory effect of its MIT domains in the absence of MIM binding would be sufficient to prevent promiscuous, non-specific interactions of all kinds of partially or fully unstructured proteins with the central pore. However, this will have to be addressed in future studies.

Relevant for the conclusions of this study is to what extent the F-peptide determines the overall conformation and leads to the strong agreement between the linear and circular peptide-bound structures. If this F-peptide binding represents a very general mode with no specific interactions, why do the extended and circular peptides all bind in exactly the same register?

Some concerns thus remain about the general advance of this study. It is true that a structure of a AAA motor with two polypeptides in the pore has not been described before. Yet, since it perfectly overlays with the previously published linear peptide-bound structure, why is this a "surprising finding for the field", especially considering that it may largely be determined by the tightly binding F-peptide? The accommodation of a second chain in the central pore is itself not unexpected. If the F-peptide sequence is indeed not "special", the authors should provide some explanation for why it stays aligned with the motor in exactly the same way for 1 or 2 strands in the pore and all structures analyzed.

The mechanistic insight of this story would for instance be significantly increased if the authors could reveal whether the motor always selectively interacts with only one strand of particular directionality and spares the other etc.

Regarding the proposed mechanism for stimulating ATP hydrolysis in subunit D, the authors propose that a subunit F-induced rotation of subunit E's small AAA domain may propagate to the arginine residues to complete coordination of ATP at the subunit D active site. However, this model of arginine engagement does in my opinion not agree with the ambiguous nucleotide density observed at the D-E interface, which was interpreted as ADP or an ADP/ADP·BeF_x_ mixture. It is expected that active sites bound to ADP vs. ATP (or ADP·BeF_x_) show significantly different arrangement and distances for the arginine fingers.

In summary, even though a AAA motor structure with two peptide chains in the central pore has not been presented before, I am not sure whether the mechanistic insight and advance of this manuscript in its current form are high enough for publication in *eLife*.

Reviewer #3:

Han et al., have adequately addressed all reviewer concerns and have made appropriate adjustments to the manuscript text and figures. Some concerns remain about the possibility that the position of the returning strand and hairpin structure in the channel may be specific to the use of these cyclic peptides and Vps4, thereby limiting the potential that this serves as a general translocation mechanism for AAA+s. Nonetheless, I agree with the authors that this work provides the first key structural insight into how AAA+s could potentially initiate from internal segments or translocate through conjugated or crosslinked sites in proteins. Thus, with these changes, I feel this work is sufficient for publication.

---

## [Author Response]

[Editors’ note: the author responses to the first round of peer review follow.]

Our decision has been reached after consultation between the reviewers. Based on these discussions and the individual reviews below, we very much regret to inform you that we cannot accommodate this work in eLife. There were no issues with the technical aspects of the findings. However, both referees were in agreement that the presented structure likely represents a specialized scenario (an initiation complex that is determined by the tightly binding F-peptide), and felt that it constitutes a variation of prior published peptide-bound structures of Vps4 which does not provide additional insights into the translocation process. There was also concern that the low complexity of the secondary strand may allow its return through the central pore without making any motor contacts, and that this may not therefore reflect a normal two-strand operation mode of AAA peptide translocases on more complex sequences. During the consultation process, the referees did note that if new conformations or substrate interactions could be identified which expand on the motor mechanism, or if an argument could be made that this the state seen here is perhaps equivalent to how Vps4 acts on Vps2 during disassembly of ESCRT-III (accompanied by appropriate functional evidence), then the impact could strengthened. We hope that you will find their comments useful in moving forward.

This summary presents three concerns, each of which is addressed below:

1) Concern #1: The presented structure likely represents a specialized scenario (an initiation complex that is determined by the tightly binding F-peptide).

There is little evidence to suggest that peptide F corresponds to an initiation site, whereas considerable evidence indicates that the Vps4-peptide F complex represents a canonical translocating conformation.

The history of identifying this peptide may have led to some confusion on this point. Phyllis Hanson’s lab showed that the human ESCRT-III proteins CHMP2A and CHMP1B contain a segment N-terminal to the MIM sequence that can associate with VPS4 (Shim et al., 2008) and promote ATPase activity (Merrill et al., 2010). We subsequently mapped a relatively tight binding sequence to a 20-residue segment of the yeast CHMP2 homolog, Vps2p (Han et al., 2015), and later found an 8-residue peptide (the F peptide sequence) that maintained similar affinity (Monroe et al., 2017). Although this path would be consistent with having identified an especially tight-binding initiation site, there are multiple reasons to think that the structures do in fact represent the canonical translocating conformation:

i) Our unpublished studies have now identified multiple unrelated peptide sequences that bind with similar affinity to peptide F, and even one that binds 30 times more strongly. Thus, despite the path that led to its identification, peptide F is not an unusually tight binding sequence.

ii) The structures show that the residues of peptide F do not make specific interactions. Instead, the structures show that very different side chains bind to a series of essentially identical binding sites (Han et al., 2017). This is exactly the sort of binding mode expected for binding and translocation of the highly diverse sequences found in ESCRT-III subunits.

iii) The peptide F sequence is not conserved in other ESCRT-III proteins that are substrates for Vps4.

iv) Finally, *all* of the other AAA ATPase peptide complex structures that have been reported bind peptide in the same manner as we reported for Vps4 (Figure 1). There are a number of points to clarify in this regard:

Reviewer 1 stated: “Many of these other structures, including the recently published substrate-bound 26S proteasome, suggest less regular, steric interactions of pore-loops with the substrate polypeptide, which may then also involve contacts with a secondary strand when substrate loops are translocated”. This is not true. Indeed, the most recent publication of a substrate-bound proteasome cryo-EM structure (Dong et al., 2019), which includes two conformational states that correspond to translocation (third mode), cites our earlier Vps4-subsrate papers in *eLife* in support of their statement: “The third mode is consistent with the proposed ATP hydrolysis mechanism of several other hexameric ATPase motors”. Moreover, overlap on both of the translocating proteasome structures of Dong et al. (PDB 6MSJ and 6MSK) shows very close agreement with the conformation and binding of peptide F by Vps4 (Figure 1). Similarly, overlap on all four of the models deposited for the stalled proteasome complex of de la Peña et al., (2018) are in close agreement with Vps4substrate coordination (Figure 1). The only deviation from a clear one-to-one match of amino acid residues occurs for one of the four del la Peña et al. structures (6EF2), where two glycines are modeled in place of one residue of the Vps4-peptide F and the other proteasome structures. This may reflect difficulty of modeling glycine conformations in extended polypeptides at worse than 4Å resolution, or it may reflect an authentic deviation from the canonical substrate conformation, as might be expected for glycine residues that lack side chains and hence engage only weakly with the AAA ATPase pore (our unpublished data show that Vps4 does not bind polyglycine).

Even structures that were modeled with the peptide bound in the opposite orientation [YME1 (Puchades et al., 2017; 6AZ0) and TRIP13 (Alferi et al., 2018; PDB 6F OX)] overlap closely with Vps4-peptide F. Although these substrate sequences run in opposite directions, their Cα atoms and side chains occupy equivalent positions with respect to the AAA pore loops (Figure 1). Because binding is mediated through side chain interactions, and because Cα atoms are superimposable when a β strand is modeled in N-to-C or C-to-N directions, the binding geometry and inferred translocation mechanisms are essentially the same as for Vps4.

Another point of confusion may be that some of the deposited structures [ClpB (Yu et al., 2018, HSP104 (Gates et al., 2017), YME1, and one of the four de la Peña et al., proteasome models] show one or two residues of the substrate in a non-β-strand conformation. Nevertheless, those structures do in fact align closely with Vps4, with the Cα atoms and side chains in the equivalent positions (Figure 1). The difference is that one or two non-glycine residues have been built with a positive phi angle in those models, which leaves residues in essentially the same place but does not conform to a β conformation. Moreover, given that positive phi angles are improbable for non-glycine residues, we suspect that these structures should be rebuilt and refined in this detail to resemble Vps4 even more closely in details of main chain conformation, in addition to the equivalent positions of side chains that are already apparent.

In short, there is no reason to think that peptide F binding to Vps4 represents a specialized scenario, and abundant reason to think that it represents a canonical translocating conformation.

**Author response image 1. respfig1:** The Vps4-peptide F complex superimposes closely with other AAA+ ATPase-substrate complexes. Overlap on AAA domains shows that pore loop 1 residues and the C α trace of the bound peptides superimpose closely. Left, side view. Right, top view. Vps4, gray. Other structures, colors: proteasome (5 structures), HSP104 (2 structures), YME1, NSF, TRIP13, ClpB (2 structures).

In order to avoid confusion over this point, the first paragraph of the Results section now includes: “Although peptide F was originally discovered as a relatively tight-binding sequence, we subsequently found that its binding affinity is comparable to a range of peptide sequences (data not shown). This indicates that its complex with Vps4 reflects a canonical translocating state, as does its structural similarity with multiple other AAA+ ATPase complexes”.

This is followed by a citation to a recent review that emphasizes the structural similarity between the Vps4 peptide complex and the other AAA+ substrate complexes.

2) Concern #2: It constitutes a variation of prior published peptide-bound structures of Vps4 which does not provide additional insights into the translocation process.

Our manuscript presents two major conceptual advances. One important new insight is the finding that hairpins can be bound, and presumably translocated, using the same mechanism as for a single strand. This is important because, as cited in our manuscript, a number of biochemical experiments [(Lee et al., 2002), (Shabek and Ciechanover, 2010), (Bodnar and Rapoport, 2017), (Burton et al., 2001)] have indicated that two (or more) polypeptide strands can be engaged in the pore of a AAA+ ATPase. Despite this long-standing interest, the mechanism by which this might happen was not indicated by the previous structural studies.

In order to emphasize this point, the final sentence of the abstract now reads:

“These observations indicate a general mechanism by which AAA+ ATPases may translocate a variety of substrates that include extended chains, hairpins, and crosslinked polypeptide chains.”

Another important conceptual advance follows from our finding that the small domain of subunit E tracks with positions seen by focused classification for subunit F. This indicates a mechanism to trigger ATP hydrolysis specifically at the active site in the final helical subunit interface. Hydrolysis at this site is a fundamental element of the sequential mechanism that has now been proposed by multiple groups, but thus far without an apparent structural basis. Thus, our observation that movement of F (coupled to completion of ATP binding at the subunit F-A interface) displaces the subunit E small domain and hence the subunit E trigger arginine residues that complete the subunit D active site, suggests an elegant structural mechanism that can be explored in future studies.

3) Concern #3: The low complexity of the secondary strand may allow its return through the central pore without making any motor contacts, and that this may not therefore reflect a normal two-strand operation mode of AAA peptide translocases on more complex sequences.

This concern is valid for one of the two structures that we reported (cF30), but it does not apply to the other structure (cFF30) described in our manuscript. The circular cFF30 peptide includes two complex “F” sequences separated from each other by 6 residues on one side and 8 residues on the other. The 6residue linker is of low complexity, but the 8-residue linker contains two lysines and a cysteine. Our structure shows the primary strand comprising an F peptide flanked by two additional residues on both sides. Thus, the configurations that most closely match the reviewers concern have a tight, reverse turn starting at the last residue (or going into the first residue) of the primary strand. As shown in Figure 2, these would still put five or six “high complexity” residues within the central region of the pore on the secondary strand. Moreover, the structure shows ample space for large side chains to be accommodated on the secondary strand of the bound cyclic peptide, and their lack of side chain density is explained by high mobility in the unconstrained environment and expectation that multiple sequence registers are represented. In short, the assertion that the secondary strand passes through the pore with low complexity residues is not correct.

**Author response image 2. respfig2:** The cFF30 peptide displays “high complexity” residues on both strands in the pore. The map shows that a peptide F sequence (red font) binds in the central region of the primary strand and that this is flanked by two ordered glycine residues on each side. The four possible configurations that display a reverse turn between the first/last ordered primary strand residue are shown. The secondary strand residues that would lie in the central region of the pore are indicated with a red box. In all cases these include at least five “high complexity” residues.

To clarify this point, the subsection “Circular peptides bind the Vps4 pore in a hairpin conformation” of the revised manuscript now includes the sentence:

“Density for these side chains is not strongly defined, although the sequence of the cFF30 peptide and the presence of two residues on either side of the primary strand F-peptide motif means that at least 5 of the returning strand residues in the pore region must have relatively large side chains.”

Reviewer #1:In this manuscript, Hill and coworkers describe the cryo-EM structure of cyclic peptide-bound Vps4 in complex with ADP·BeF_x_ and the VSL domain of Vta1, suggesting how this AAA+ motor may translocate loops or two strands of protein substrates. The authors used circularized 30-mer peptides with one or two copies of the 8-residue segment DEIVNKVL (F peptide) that was derived from the ESCRT-II subunit Vps2, specifically binds to the Vps4 pore, and whose structure bound to Vps4 has previously been solved by the same group. This primary segment of the circular peptides was found to bind indistinguishably from the linear peptide, whereas the secondary segment forms β-ladder interactions with the primary segment and passes through the Vps4 pore without making any significant contacts.The presented data are of good quality and provide insights into how Vsp4 may simultaneously translocate two polypeptide chains. However, the strong agreement with the previously published structure of linear peptide-bound Vps4 limits the extent of conceptually advances.

As noted above, an important conceptual advance is that two strands can bind, and presumably be translocated, using the same mechanism as employed for a linear-bound peptide. This insight was not anticipated in previous publications.

Due to the use of the specifically-binding F-peptide combined with a low-complexity (G, A, and S-rich) secondary segment, it is not too surprising that affinities and peptide-motor interactions are highly similar or identical to those previously described.

As discussed under concern #3, above, the secondary segment is not low complexity in the case of peptide cFF30.

The authors strongly generalize their findings and suggest throughout the manuscript, including in the title and abstract, that other AAA+ ATPases may process substrate loops or multiple chains in a very similar manner. However, it is unclear to what extent the observed conformations and interactions are relevant for other AAA+ motors, as the presented structure likely reflects an initiation complex in which a specifically recognized peptide binds with micro-molar affinity to a static, ATP-hydrolysis-deficient Vps4. As acknowledged by the authors, other substrate-bound AAA+ motors do not show peptides in the same β-strand conformation with highly ordered subunit contacts, which also makes it unlikely that the secondary strand of a substrate loop forms similar β-ladder interactions as observed here for Vps4. Many of these other structures, including the recently published substrate-bound 26S proteasome, suggest less regular, steric interactions of pore-loops with the substrate polypeptide, which may then also involve contacts with a secondary strand when substrate loops are translocated.

As discussed under concern #1, above, it seems unlikely that our structure represents an initiation complex. Moreover, our structure is in fact very similar to essentially all other substrate-bound AAA+ motors, and that deviations in the other substrate complexes likely represent sloppy model building (positive phi angles) or special cases such as localized glycine residues. As indicated under Concern #1, the point has been clarified in the revised manuscript by adjusting wording in the first paragraph of the Results section and Discussion section and by adding a citation to a review that was published since our original submission.

The authors favor a model in which the primary strand binds in C-to-N terminal orientation to the Vps4 subunits A – E, with the secondary strand spared from any interactions. This model would imply that folded domains N-terminal of the motor-bound primary strand would get preferentially unfolded. Biochemical studies of substrate processing by the 26S proteasome (e.g. Piwko and Jentsch, 2006) indicate that initiation on an internal loop can lead to processing of the N-terminal, C-terminal, or both segments, speaking against selective interactions with only the primary strand. It is thus questionable whether the model presented here applies to other AAA+ motors including the proteasome.

We do favor one orientation of peptide binding for Vps4, but our manuscript makes it clear that we have are unable to definitively determine the orientation of binding from the current data. For example, the “Circular peptides bind the Vps4 pore in a hairpin conformation” section includes the sentences: “The map-model correlation coefficients and the EMRinger scores (Barad et al., 2015) all slightly favor the assigned orientation, but are not definitive. This ambiguity is expected for the current 3.6 Å resolution, and the peptide orientation remains an important question for future studies.”

We also make it clear that other AAA+ ATPases appear to bind substrate in the opposite orientation and that the same mechanism can apply in either orientation. For example, the subsection “Implications for mechanism and function” now includes:

“This orientation is consistent with the biological role of Vps4 in translocating toward the ESCRT-III N-terminal domain, but other AAA+ ATPases apparently translocate their protein substrates in the opposite direction (N-to-C) (Alfieri et al., 2018; Puchades et al., 2017) or in either direction (Augustyniak and Kay, 2018). Indeed, the same mechanism of translocation could be applied to substrates bound with their primary strand in either orientation because side chains make a major contribution to binding, and forward and reversed β-strands can superimpose their Cα atoms and side chains.”

The authors suggest (or imply through their wording) that they identified a new mechanisms of substrate engagement: "Our data indicate a third potential mechanism in which substrate engagement and translocation initiate from an internal segment by binding of a folded hairpin directly within the hexamer pore." Even though structural data have so far been missing, internal initiation and the translocation of multiple chains had already been well established by extensive biochemical studies, not only for the 26S proteasome, but also for members of the Clp family. These previous studies should be cited and the presented Vps4 structure discussed accordingly.

This is a good point. Accordingly, we have deleted discussion of initiation of translation from an internal loop in the revised manuscript. As suggested, we have retained the discussion of implications for translation of more than one polypeptide chain at a time, and have cited literature reporting this activity of the 26S proteasome, ClpX, and Cdc48.

Regarding the coupling of ATP hydrolysis and substrate translocation, a similar coordination of conformational changes as described here for subunits F and E has also been observed for subunits of the substrate-engaged 26S proteasome in various stages of the ATPase cycle (de la Pena et al., 2018) and should be discussed.

The beautiful paper by de la Pena et al., is cited in the first paragraph of the Introduction, where the similarity between this proteasome structure(s) and structures of other AAA+ ATPase complexes with substrate peptides is used to set up the argument in favor of the sequential, hand-over-hand, conveyer-belt model. The de la Pena et al. paper is also cited in the section “Comparison with other AAA+ ATPase-peptide complexes”. The de la Pena et al., paper is not cited, however, in the section “Insights to coupling of ATPase activity and substrate translocation”. This is because this section focuses on the question of how ATP hydrolysis is selectively triggered at the subunit D active site given that the subunit A, B, C, and D active sites appear to be essentially identical. Our Monroe et al., 2017 paper was the first to propose subunit D as the site of ATP hydrolysis and, indeed, the directionality of translocation seems to make selective hydrolysis at this site a mechanistic requirement. While the de la Pena et al. paper provides multiple important insights, at our reading it does not seem to address the question of how hydrolysis is selectively favored at the site equivalent to Vps4 subunit D.

The presented structures add some details, but no groundbreaking new findings about the interaction of Vps4 with the VSL domain of the Vta1 activator.

We agree that these details are not as important as the finding that two polypeptides can bind the pore in the same manner as a single extended chain, and by implication can be translocated using the same mechanism. Nevertheless, they do advance understanding and merit inclusion in the context of the whole manuscript, as is done quite briefly in our revised manuscript.

In summary, even though it is interesting to visualize Vps4 bound to a circular peptide, conceptually new findings are somewhat limited due to the strong similarities to the previously published structures of the linear peptide-bound motor. Since the presented structures likely represent an initiation complex with a tightly binding peptide, it remains unclear to what extent observed interactions apply to processively translocating Vps4 or even other AAA+ motors, especially in light of existing substrate-bound structures that already show deviations in peptide conformations and orientations within the central pore. The authors should try to focus their manuscript primarily on Vps4 and reduce their generalization about the mechanisms of other AAA+ motors. Contrary to the author's suggestion, these structures did not identify a novel mechanism for internal initiation of substrate processing, however, they do represent the first visualization of such a complex. I'll have to leave it to the reviewing editor to decide whether the presented advances are significant enough to consider publication in eLife.

This is a clear summary of reviewer #1 concerns, each of which was addressed individually above.

Reviewer #2:In this manuscript Han, et al. determined a 3.6-Å structure of the Vps4 AAA+ ATPase, which is essential for dissociating ESCRT complexes, bound to a cyclic peptide containing the Vps2 binding sequence. The significant results are that the peptide is bound in a β-ladder hairpin conformation with two strands spanning the Vps4 translocation channel. The Vps2 sequence is bound in the same arrangement with defined pore-loop contacts as previous structures while the returning strand is more flexible and runs along the helical interface without making contact with the hexamer. The structure is important because it shows that two strands can be accommodated in the translocation channel; all previous structures of AAA+ translocases show a single unfolded strand in the channel. The authors model this β-hairpin into these previous structures and propose this as evidence that related AAA+s can initiate and translocate internal loops of substrates in addition accessible termini – although this is tenuous given that no experiments directly address this functionally and other AAA+s are not tested. Additionally, the authors are able to increase the occupancy of the Vta1 cofactor subunit and thereby establish its stoichiometry in the complex and improve the reconstruction and modeling of this region compared to previous structures. The work in this manuscript is technically very sound with compelling substrate binding analyses, cryo-EM structure determination and molecular modeling methods.

We agree with this summary of the primary findings in our manuscript. We also agree that the modeling with other AAA+ ATPase is necessarily speculative, although we are more confident in the relevance of this comparative analysis than the reviewer, whose concerns may be influenced by misunderstanding about the nature of the Vps4 complexes. As discussed elsewhere, the Vps4 complexes do not appear to be specialized initiation states, but are instead highly representative of translocating complexes of AAA+ ATPases in general.

However, two points of concern regarding the functional significance and novelty of these results reduce enthusiasm for publication in eLife: (1) These results may be a specific consequence of the experimental setup which includes the use of a cyclic peptide with low-complexity Gly, Ala and Ser residues outside Vps2 that may be required to fit in the channel, use of ADP·BeF_x_, glutaraldehyde crosslinking, and a truncated Vps4 with the VSL domain added for stability. Thus, it is unclear whether a native Vps4 complex could bind the full-length Vps2 substrate by this mechanism or if other AAA+s can translocate multiple strands through the channel.

The issues raised in this paragraph are each addressed separately:

(i) Use of a cyclic peptide with low-complexity Gly, Ala and Ser residues outside of Vps2 that may be required to fit in the channel.

As discussed above under Concern #3, this concern is not valid for the cFF30 peptide whose binding affinities and Vps4 complex structure are reported in our manuscript.

(ii) Use of ADP·BeF_x_.

We don’t agree that this should be a concern. It is usual for nonhydrolyzable analogs such as ATPγS, AMPPNP and ADP·BeF_x_ to be used to visualize the active state of ATPases. If not inhibited with a non-hydrolyzeable nucleotide, ATPases are almost invariably inhibited by mutagenesis or by some other inhibitor (such as inhibition of deubiquitylation by the proteasome. It is curious that we observe peptide binding in the presence of ADP·BeF_x_ or ADP·AlF_x_ but not in the presence of ATPγS or AMPPNP (Han et al., 2015). However, a good explanation is provided in the same Han et al. 2015 study, which found that ADP·BeF_x_ and ADP·AlF_x_ are more effective at stabilizing the Vps4 hexamer. This implies that ADP metal fluorides are better mimics of ATP in the Vps4 active site than ATPγS or AMPPNP, which is quite reasonable give the extensive coordination of phosphates in Vps4 active site.

(iii) Glutaraldehyde crosslinking.

The use of glutaraldehyde crosslinking is commonly employed in many cryo-EM structure determinations to protect the complex against disruptive interactions with the air interface immediately prior to vitrification. Importantly, our structure determination of Vps4 in the presence of glutaraldehyde is very similar to structures of Vps4 determined in the absence of glutaraldehyde (Su et al., 2017 and Sun et al., 2017) of intact particles.

(iv) Truncated Vps4 with the VSL domain added for stability.

This is not a concern because structures of full length Vps4 in the presence and absence of the VSL domain are essentially identical to our structure (Su et al., 2017; Sun et al., 2017).

(v) It is unclear whether a native Vps4 complex could bind the full-length Vps2 substrate by this mechanism.

We agree that it is unclear whether or not Vps4 binds full-length Vps2 with two strands within the translocation pore. We do not intend for that to be taken as a conclusion of this study, and have clarified this point by modifying the subsection “Comparison with other AAA+ ATPase-peptide complexes” to include the text:”Thus, regardless of whether or not Vps4 binds its substrates in a hairpin conformation in vivo, it seem likely that at least some of the AAA+ ATPases may use the same mechanism to translocate linear polypeptides and more complex substrates, such as protein loops, crosslinked substrates, and ubiquitin adducts, as have been indicated for the proteasome (Kraut and Matouschek, 2011; Lee et al., 2002; Shabek and Ciechanover, 2010), CDC48 (Bodnar and Rapoport, 2017), and ClpX (Burton et al., 2001).

(vi) *… or if other AAA+s can translocate multiple strands through the channel.*

Biochemical studies have already established that other AAA+ ATPase can translocate multiple strands through the channel. These studies are cited in our manuscript for the proteasome (Lee et al., 2002; Shabek and Ciechanover, 2010; Kraut and Matouschek 2011), ClpX (Burton et al., 2001), and Cdc48 (Bodnar and Rapoport 2017). A key point of our paper, therefore, is not that Vps4 does translocate a hairpin structure for any specific substrate, but rather that

Vps4 and presumably other AAA+ ATPases appear to have the potential to translocate two-strand substrates using the same mechanism as inferred for single strand substrates.

2) The Vps4 hexamer and bound Vps2 peptide are in an identical configuration and nucleotide state as the two previously published structures (one at higher resolution) by this group and the second strand of the substrate is flexible and passively bound without any specific contacts with the Vps4 channel, thus the insight into the AAA+ translocation mechanism, that Vps4 can accommodate two strands under these conditions, has modest impact to the field.

We believe that the observation that two strands can bind and (presumably) be processed in the same way as a single strand is a surprising finding for the field. The point is not that a different mechanism of translocation has been found, but that the same mechanism likely applies to a substantially different substrate.

Specific comments:-Were other nucleotides tested for binding to the cyclic peptide? Hydrolysable nucleotides, such as ATP or ATPyS, would be worth testing to potentially capture different nucleotide states of the subunits or different translocation intermediates.

We did not test different nucleotides in this study. We previously surveyed binding of linear peptides in the presence of multiple different nucleotides (ADP, ATP, ATPγS, AMPPNP, ADP·BeF_x_, and ADP·BeF_x_), and only detected binding to ADP·AlF_x_ and ADP·BeF_x_ (Han et al., 2015). The absence of binding to ATP makes sense because, unlike a number of the other recently reported AAA+ ATPase structures, the enzyme we are using has the wild type sequence and an intact active site. Therefore, as soon as a short peptide binds in the presence of ATP is expected to be translocated off the enzyme. The absence of binding to the other nucleotides likely reflects the lower stability of the Vps4 hexamer that we see in the presence of those nucleotides as judged by size exclusion chromatography (Han et al., 2015). Our preferred interpretation, therefore, is that ADP·AlF_x_ and ADP·BeF_x_ are better mimics of non-hydrolysable ATP at the Vps4 active sites. It is possible that altered biochemical conditions would reveal biochemical binding of peptide in the presence of different nucleotides. But we don’t consider that possibility to be of sufficient importance to merit further study at this time, especially because our structure of the Vps4 complex resembles structures of multiple other AAA+ ATPases (including Vps4; Su et al., 2017; Sun et al., 2017) determined in a variety of active, inactive, and nucleotide-bound states (Han and Hill, 2019).

In order to reduce potential confusion over this issue, the first paragraph of the Results section in the revised manuscript now includes “ADP·BeF_x_ was used as the non-hydrolysable ATP analog because our earlier studies indicated that it stabilizes the Vps4 hexamer and supports peptide binding to a greater extent than ADPPNP or ATPγS, presumably because it is a better mimic of ATP at the Vps4 active site”

-What is the solution structure of the cyclic peptide? Does it form the β-hairpin structure or is this a consequence of Vps4 binding? Perhaps CD spectra of the peptide could be measured.

We do not believe that this proposed experiment would provide an important insight. It seems likely that the cyclic peptide adopts a variety of conformations in solution, include those with some β-hairpin character. And it seems inevitable that interaction with Vps4 further defines the peptide conformation. Importantly, the β-hairpin conformation is an energetically accessible conformation for essentially all sequences.

-Were other cyclic peptides tested with difference sequences for the hairpin? It seems that this two-stranded β-ladder complex may require low complexity or specific amino acids with minimal side chains for the returning strand that is adjacent the Vps2 sequence in order to fit in the channel. Functional significance would be improved if a solution-state β-sheet or the full α-helix that contains the Vps2 sequence were tested.

Two different peptides are presented in our manuscript, one of which (cF30) has an extended stretch of low complexity residues, the other of which (cFF30) avoids an extended stretch of low complexity residues specifically to mitigate this concern. We do not believe it would be highly informative to survey a few more individual sequences. Rather, now that we have established the possibility of hairpin structures binding in the same manner as single chains, our focus will be on determining whether or not this can and does occur with the authentic substrates of a variety of AAA+ ATPases. Although resolving that important question is beyond the scope of the current manuscript.

-The occupancy and nucleotide state of the subunits is discussed, but no data is shown in support of this.

We did not emphasize this in the submitted manuscript because the observations are essentially identical to our findings with Vps4 in complex with a linear peptide (Han et al., 2017). As requested, we now show the relevant data by including figures of density at the nucleotide sites as Figure 2C and the associated supplemental video in the revised manuscript.

-In their translocation model the authors propose that subunit F, which is disconnected from the substrate and asymmetric with respect to the helical arrangement of the hexamer, moves ~30 Å to the subunit-A end of the helix during a translocation step. By focussed classification of this region of the hexamer they identify different positions of subunit F. Local resolution for this region is stated to be 4-7 Å for these classes, however no data is shown. How was the focussed classification performed? Is the cryo-EM density for F improved by focussed classification? Only the Vta1 focussed classification is discussed in the methods. From Figure 4B it is difficult to tell how these positions/conformations of F fit with their translocation model or that "they span a substantial fraction of the path" that is proposed for a translocation step. Please show the range of motion and relative positions of subunit A and E along the channel axis.

The Materials and methods section has been updated to describe how the subunit F data were processed. Furthermore, the revised manuscript includes new figures that show how focused classification over subunit F was performed (Figure 2—figure supplement 4) and validation of the results (Figure 2—figure supplement 5). This includes local resolution heat maps, and gold standard, corrected FSC plots. We also show the structural relationship of F_1_, F_2_, and F_3_ to each other (Figure 2—figure supplement 6), their variation in hinge angle (Figure 5A), and their relationship to focused classification models of subunit F from other Vps4 datsets along the proposed transition of subunit F during the translocation cycle (Figure 5C).

-As rationale for a potential conserved function of AAA+s binding a two-strand β-ladder structure the authors make claims that are questionable and not referenced. It is stated "AAA+ ATPases are often thought to initiate substrate engagement from a protein terminus". Please provide supporting studies for this claim. For AAA+ disaggregases it has been proposed for a number of years that engagement can occur from internal segments (see Haslberger et al., 2008).

This is a good point. We have removed the claim that AAA+ ATPases are thought to initiate from a protein terminus from the revised manuscript.

Additionally it is stated that AAA+ hexamerization could occur "around a linear portion of their substrate", but "this possibility is relevant for family members that show inherently weak hexamerization in the absence of bound substrate but is unlikely to be applicable for robust hexamers like the proteasome and Hsp104". This claim is inaccurate, and no supporting references are provided. In fact, the "lock washer" helical conformation has only been observed in the presence of substrate for these AAA+s. Crystal structures show a continuous helix, hexamerization is dynamic and dependent on nucleotide state (see DeSantis et al., 2012, Aguado et al., 2015 and Uchihashi et al., 2018) and the substrate-free cryo-EM structures of VAT, ClpB and Hsp104 show an open helical spiral that is an entirely different arrangement and likely incompatible with translocation. Thus, while hexamerization/oligomerization can occur without substrate, these complexes are highly dynamic. Therefore, hexamerization around the substrate or, more likely, passing an internal segment though the seam interface are highly plausible models for engagement of internal segments by the proteasome, Hsp104 and other AAA+s.

We have removed the qualifier about robust hexamers from the revised manuscript.

[Editors’ note: the author responses to the re-review follow.]

Reviewer #2:I appreciate the authors' further discussion about the F-peptide conformation and its similarities to structures observed for related AAA motors. I agree that the overall arrangement and mode of peptide interaction is similar to other motors, as this is largely dictated by the very consistent helical-staircase arrangement of ATPase subunits. The question is about the high regularity of those interactions in Vps4, whether substrates in general have to adopt a defined β structure in the pore, and to what extent the observed conformation originates from the tight binding (K_D_ = 250 nM) and consequent potential energy minimum of the characterized state. A comparison with peptide conformations in other AAA motors is certainly warranted, but needs careful phrasing, especially because the intermediate resolution of those structures makes an assessment of phi/psi angles difficult. Even for the Vps4 structure presented here, the authors' strong claims about β strand conformation is oddly contrasted by their uncertainty about N-to-C or C-to-N directionality of the bound peptide.

The revised text notes the observed similarity between reported structures, why we favor a canonical β conformation for bound substrate, the limits of the current structural models, and the likelihood that some variation from the standard β conformation. Specifically, we have included the following text in the introduction: “These AAA+ ATPase complexes bind the substrate polypeptide in the central pore in an extended conformation, which in the case of Vps4 has been modeled as a β-strand-like conformation whose right-handed helical symmetry (60° rotation and ~6.5 Å displacement every two amino acid residues) matches the symmetry of the helical AAA+ ATPase subunits (Han et al., 2017; Monroe et al., 2017). Although the resolution of currently available AAA+ ATPase substrate complexes makes it challenging to model precise details of the substrate structure, this conformation is appealing because it allows the substrate to bind the helical AAA+ ATPase subunits with successive dipeptides of the substrate making equivalent interactions with the enzyme and because it is accessible for almost all amino acid residues. Some variation from the canonical conformation is likely to occur, especially for sequences that contain proline, which has a fixed -60° phi angle, and glycine, which is flexible and lacks a side chain, which seems to be an important for binding.”.

The reason we can make stronger claims about β conformation than about peptide directionality is two-fold. Primarily, it is much easier to tell from medium resolution maps whether or not a segment is in a β conformation than its direction. Unlike helices, for which the direction of the side chain gives a clear indication of directionality, strands lack any such indicator. Indeed, at medium resolution the density for β strands is almost exactly superimposable in N to C vs C to N directions. The distinction only becomes apparent when the resolution is sufficient to resolve main chain carbonyl groups. Second, in all of the reported structures, the AAA+ ATPase pore loops that mediate binding display the same symmetry as a β strand, thereby providing a chemical rationale for why substrate should bind in a β conformation. No equivalent rationale is strongly apparent for a preferred orientation.

Independent of whether the F-peptide complex is indeed an initiation complex, its high affinity is a bit surprising given the non-specific nature of interactions during substrate translocation. If peptides in general bound with similarly high affinity, as suggested by the authors, I am wondering how Vps4 can maintain a high enough selectivity, and whether the autoinhibitory effect of its MIT domains in the absence of MIM binding would be sufficient to prevent promiscuous, non-specific interactions of all kinds of partially or fully unstructured proteins with the central pore. However, this will have to be addressed in future studies.

We note that Vps4 selectivity is currently understood to be maintained by (at least) two mechanisms: MIT autoinhibition (as noted by the review) and assembly. In the former case, our published biochemical studies (Han et al., JBC, 2015) have demonstrated that MIT-mediated autoinhibition is sufficient to block binding of peptides that have comparable affinity to peptide F. In the latter case, binding of the Vps4 N-terminal MIT domain to MIM motifs on ESCRT-III subunits recruits Vps4 to high local concentration at filaments of ESCRT-III subunits. This high local concentration, coupled with the associated high local concentration of the Vta1 cofactor protein, promotes Vps4 assembly to the active hexameric state only when the enzyme is associated with the ESCRT-III substrate.

Relevant for the conclusions of this study is to what extent the F-peptide determines the overall conformation and leads to the strong agreement between the linear and circular peptide-bound structures. If this F-peptide binding represents a very general mode with no specific interactions, why do the extended and circular peptides all bind in exactly the same register?

Register is not rigorously defined because, as acknowledged in the text, we can’t confidently determine whether the peptide binds in the N to C or the C to N directions.

Nevertheless, we can see that all of peptide F is ordered in the previously reported structures with the linear 8-residue peptide. We are confident that this binds in the register modeled – or its reverse or a mixture of both orientations – because the density has the right length to accommodate 8 residues and because the structure presents 8 side chain binding pockets. Alternative registers are presumably avoided because they would lose the binding energy associated with occupying one or more of the 8 available binding pockets.

Similarly, the cF30 and cFF30 peptides were designed with two glycine residues flanking each of the 8-residue peptide F sequences. Because polyglycine binding to Vps4 is undetectable – which is still unpublished but is anticipated from the structure – the same argument applies: a shift in register will be disfavored because it would result in the loss of favorable interactions with a substrate side chain.

Some concerns thus remain about the general advance of this study. It is true that a structure of a AAA motor with two polypeptides in the pore has not been described before. Yet, since it perfectly overlays with the previously published linear peptide-bound structure, why is this a "surprising finding for the field", especially considering that it may largely be determined by the tightly binding F-peptide?

The revised manuscript does not describe any of the findings as being surprising or unexpected, etc. Nevertheless, we believe it is important because there is clear evidence in multiple systems that AAA ATPases must be able to accommodate two different polypeptide strands (e.g., as when translocating from the middle of a polypeptide substrate or when translocating ubiquitinated substrates), and the structure provides the first visualization of how two polypeptide strands can be accommodated within the central pore of the hexamer.

The accommodation of a second chain in the central pore is itself not unexpected. If the F-peptide sequence is indeed not "special", the authors should provide some explanation for why it stays aligned with the motor in exactly the same way for 1 or 2 strands in the pore and all structures analyzed.

The reason for conserved alignment/register is explained above. Peptide F is “special” in the sense that in the contexts we have studied it is either always flanked by glycines, which do not bind Vps4 detectably, or it exists on its own as just the 8-residue peptide.

The mechanistic insight of this story would for instance be significantly increased if the authors could reveal whether the motor always selectively interacts with only one strand of particular directionality and spares the other etc.

We agree that the question of directionality is very interesting, and we are actively pursuing it at this time. However, it is not trivial to resolve at the currently available resolution, and is beyond the scope of the current study.

Regarding the proposed mechanism for stimulating ATP hydrolysis in subunit D, the authors propose that a subunit F-induced rotation of subunit E's small AAA domain may propagate to the arginine residues to complete coordination of ATP at the subunit D active site. However, this model of arginine engagement does in my opinion not agree with the ambiguous nucleotide density observed at the D-E interface, which was interpreted as ADP or an ADP/ADP·BeF_x_ mixture. It is expected that active sites bound to ADP vs. ATP (or ADP·BeF_x_) show significantly different arrangement and distances for the arginine fingers.

The ambiguous nature of nucleotide density at the subunit D active site is an observation (albeit one that is not very satisfying!). This does not alter the facts that the finger arginines are important for ATP binding, that their displacement will be tightly coupled to the hydrolysis reaction, and that apparent movement of subunit F is coupled to finger arginines at the subunit D active site through correlated motion of the subunit E small ATPase domain.

In summary, even though a AAA motor structure with two peptide chains in the central pore has not been presented before, I am not sure whether the mechanistic insight and advance of this manuscript in its current form are high enough for publication in eLife.

The question of impact inevitably has a subjective element. We appreciate the thoughtful feedback.

Reviewer #3:Han et al., have adequately addressed all reviewer concerns and have made appropriate adjustments to the manuscript text and figures. Some concerns remain about the possibility that the position of the returning strand and hairpin structure in the channel may be specific to the use of these cyclic peptides and Vps4, thereby limiting the potential that this serves as a general translocation mechanism for AAA+s. Nonetheless, I agree with the authors that this work provides the first key structural insight into how AAA+s could potentially initiate from internal segments or translocate through conjugated or crosslinked sites in proteins. Thus, with these changes, I feel this work is sufficient for publication.

Thank you for the positive comments and the assessment that this work is sufficient for publication.